# Wanting what hurts: D1 dopamine neuronal stimulation in CeA is sufficient to induce maladaptive attraction
David Nguyen ✉ & Kent C. Berridge ✉

Maladaptive desires, such as addictions, can arise and persist even when their outcome is not enjoyed. A laboratory prototype of maladaptive desire is 'wanting what hurts'. This has been produced in rats by pairing optogenetic stimulations of neurons in central nucleus of amygdala (CeA) with voluntary encounters of an electrified shock rod. However, it remains unknown which particular CeA neuronal subtypes can generate maladaptive attraction. Here we used Cre targeted optogenetic stimulation to assess relative contributions of CeA neuronal subtypes that express D1 dopamine receptors, D2 dopamine receptors, or CRF as neurotransmitter. We report that selective stimulation of D1-expressing CeA neurons is sufficient to induce shock rod attraction, similar to that produced by hSyn-targeted stimulation of general CeA neuronal populations. Both caused similar levels of self-administered shocks. CeA D1 rats and CeA hSyn rats were also motivated to overcome a barrier to reach the shock rod, and to seek out Pavlovian cues associated with shocks from the shock rod. These features, plus mesolimbic Fos activation, indicated that maladaptive attraction was mediated by incentive motivation mechanisms usually reserved for rewards. Our results reveal a special role for D1-expressing CeA neurons in producing 'wanting what hurts' as a prototype of addictive-like motivation.

Amygdala-related circuitry helps attribute motivational significance to stimuli associated with positive rewards, as well as to stimuli associated with negative threats[1–5]. The central nucleus of amygdala (CeA) in particular is a striatal-type nucleus in macrosystem frameworks[6], which is largely GABAergic and receives mesolimbic dopamine projections and contains neurons that can express either D1-type or D2-type dopamine receptors[7]. In keeping with CeA's striatal-type status, local neurobiological stimulations within CeA can generate intense appetitive motivation for food, sex, drugs of abuse, or other rewards[8–14].

Previously it was reported that maladaptive attraction to an electrified shock rod could be induced in rats by associatively pairing brief optogenetic channelrhodopsin stimulations of CeA neurons (CeA ChR2) with voluntary encounters of the electrified metal shock rod[12]. Consequently, CeA ChR2 rats approached and hovered closely over the shock rod and repeatedly touched it with paw, nose, or mouth despite receiving many noxious electric shocks[12]. CeA ChR2 rats also were sufficiently motivated to repeatedly climb over a large obstacle to reach and touch the shock rod, and sought out Pavlovian CS + cues associated with rod shocks, as if the shock-associated CS+s were reward cues[12]. When CeA ChR2 stimulations were paired with actual rewards for other rats, such as either earning intravenous cocaine or sucrose, rats whose CeA ChR2 stimulation was selectively paired with sucrose (but

not cocaine) became 'sucrose addicts': intensely pursuing and consuming only sucrose, while ignoring intravenous cocaine. Other rats whose CeA ChR2 stimulation was paired with cocaine (but not sucrose) became 'cocaine addicts': selectively pursuing only cocaine while ignoring sucrose[12]. Yet, most CeA ChR2 rats failed to robustly self-stimulate CeA laser by itself when given the opportunity without shocks, despite being strongly attracted to their laser-paired shock rod, sucrose, or cocaine. This pattern led to the hypothesis that CeA ChR2 laser-pairing can transform the incentive value of a paired unconditioned stimulus to make it intensely 'wanted', rather than simply transferring laser reward value to that stimulus[12].

'Wanting what hurts' illustrates a tenet of the incentive sensitization theory of addiction, namely that incentive motivational 'wanting' can become independent of hedonic 'liking' for the same target via activation of mesocorticolimbic circuitry[15]. In support, the CeA ChR2 attractions to shock rod, sucrose or cocaine all similarly recruited neurobiological activations in distant mesocorticolimbic circuitry, including ventral tegmentum (VTA), nucleus accumbens (NAc) and related limbic structures[12].

CeA contains multiple neuronal subtypes, including neurons that express D1 dopamine receptors, other neurons that express D2 receptors, some that co-release corticotropin releasing factor (CRF) as a neurotransmitter, etc. Which CeA neuronal subtypes are particularly capable of

Department of Psychology, University of Michigan, Ann Arbor, MI, USA. ✉e-mail: davnguye@umich.edu; berridge@umich.edu

generating maladaptive 'wanting what hurts'? Here we compared Cre-targeted CeA ChR2 laser pairings with shock rod encounters in transgenic strains of D1 Cre, A2(D2) Cre, and *Crh* Cre rats. Our aim was to assess the respective roles of CeA neurons that either express D1-type dopamine receptors, CeA neurons that express D2-type dopamine receptors and A2 adenosine receptors, and neurons that express CRF as neurotransmitter. Finally, we compared these selective neuronal stimulations with effects of hSyn targeted general CeA ChR2 stimulation of all neuronal subtypes. We report that D1 neuronal stimulation in CeA is sufficient to generate maladaptive attraction to a shock rod at levels that are comparable to hSyn stimulation of all CeA neuronal subtypes. D1 and hSyn rats were also motivated to overcome an obstacle to reach the shock rod, sought Pavlovian cues associated with the shock rod, and showed mesolimbic Fos activation patterns that are indicative of incentive motivation.

## Results

Results indicated within-group variation to be small in shock rod tests, but large in laser self-stimulation tests. That is, some individual rats in hSyn ChR2 and D1 ChR2 groups self-stimulated CeA laser either intensely or moderately, whereas other rats in the same groups failed to self-stimulate laser at all. With such individual differences in mind and in accord with a reviewer's suggestion, laser self-stimulation data are presented first, and shock rod attraction data are then presented separately for individuals that did self-stimulate laser versus individuals that failed completely. Furthermore, in all following analyses, there were no sex differences unless otherwise stated.

### CeA laser self-stimulation

As a group, hSyn ChR2 rats self-stimulated significantly more on the active porthole/spout ($267.3 \pm 89.28$; mean $\pm$ SEM) than on the inactive porthole/spout that earned nothing ($10.96 \pm 2.36$) across days 1–3 (paired-samples $t$-test, $t_{18} = 2.91$, $p = 0.009$; Wilcoxon Signed-Rank test, $W = 2.94$, $p = 0.003$). However, only about 60% of hSyn ChR2 individuals actually self-stimulated, as others did not at all (Fig. 1a, b).

Similarly, D1 ChR2 rats as a group did significantly self-stimulate laser, making more active responses ($607.10 \pm 248.43$) than inactive responses ($11.98 \pm 1.82$) across the 3 days (paired samples $t$-test, $t_{15} = 2.4$, $p = 0.03$; Wilcoxon Signed-Rank test, $W = 3.36$, $p = 0.001$). But again, only about 60% of D1 ChR2 individuals contributed to this effect, whereas other D1 ChR2 rats completely failed to self-stimulate (Fig. 1c, d).

A2(D2) ChR2 rats failed to self-stimulate, and instead slightly avoided stimulation of CeA A2(D2) neurons, making fewer active ($1.22 \pm 0.22$) than inactive ($2 \pm 0.19$) responses (paired samples $t$-test, $t_2 = 6.95$, $p = 0.02$) (Fig. 1e).

CRF ChR2 rats overall failed to self stimulate, and did not differ in active versus inactive responses across all 3 days (paired samples $t$-test, $t_8 = 0.96$, $p = 0.366$) (Fig. 1f). Control eYFP rats as a group also failed to show laser self-stimulation, and did not make significantly more active responses ($13.83 \pm 5.44$) than inactive responses ($12.63 \pm 4.73$), (paired samples $t$-test, $t_7 = 0.16$, $p = 0.88$) (Fig. 1g) (see Supplementary Note 1 for further analyses).

### Paired hSyn ChR2 and D1 ChR2 stimulation in CeA creates attraction to shock rod

**Control rats: shock rod avoidance and defensive behavior.** Control eYFP rats touched the shock rod (Fig. 2a) only a few times in initial exploration on the first day, receiving $3.2 \pm 0.55$ shocks. They diminished further to $1.45 \pm 0.32$ shocks on the second day, and $0.6 \pm 0.15$ shocks on the third day (Fig. 2b). Control eYFP rats also exhibited an anti-predator behavior called defensive burying toward the shock rod: using forepaws to kick cob bedding towards the rod and potentially bury it[16], spending $128.85 \pm 41.02$ s on average engaged in defensive burying on day 1, $6.02 \pm 3.82$ s on day 2, and $3.84 \pm 3.84$ s on day 3. That is, as days went on control eYFP rats began to simply avoid the shock rod, and stayed on the opposite side of the chamber, so that active defensive burying behavior declined as no more shocks occurred.

**hSyn ChR2 rats: attraction to shock rod.** The hSyn ChR2 rats touched the shock rod frequently and received $16.12 \pm 1.12$ shocks on day 1. On Day 2, hSyn ChR2 rats received $12.31 \pm 1.55$ total shocks, and on Day 3 received $13.31 \pm 1.47$ shocks (Fig. 2b). We explicitly compared shock rod attraction in individuals that went on to self-stimulate laser in the nose-poke/spout-touch task, versus individuals that failed to self-stimulate (Fig. 2f): hSyn ChR2 rats that did self-stimulate laser made slightly more shock rod contacts ($16.86 \pm 4.49$) than hSyn ChR2 rats that failed to self-stimulate ($14.47 \pm 5.2$). However, the number of shock rod contacts made did not significantly differ between the two groups (independent-samples $t$-test, $t_{17} = 1.06$, $p = 0.31$) (Fig. 2g), and there was no correlation between laser self-stimulation score and shock-rod attraction for hSyn ChR2 rats (Pearson's correlation, $r_{18} = 0.35$, $p = 0.14$). Thus, while motivation to

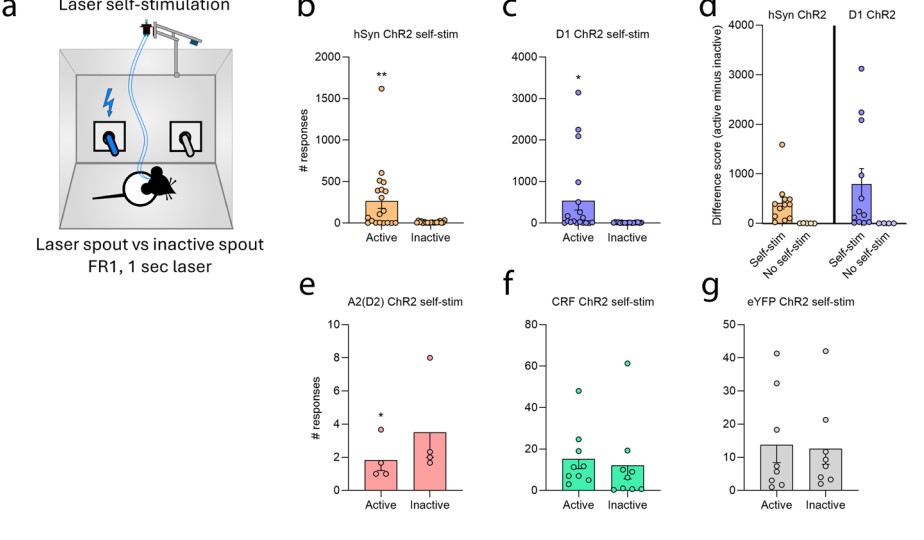

**Fig. 1 | Laser self-stimulation. a** Diagram depicts the CeA laser self-stimulation task in which rats could earn 1-sec CeA laser illuminations by touching a designated active empty metal spout or by making nosepokes into a designated active porthole. Contacts on an alternative inactive porthole/spout earned nothing. **b** Both hSyn ChR2 ($N = 19$) and **c** D1 ChR2 ($N = 18$) rats self-stimulated laser as entire groups. However, marked individual differences were observed, as a few rats self-stimulated at very high rates, but a number of other D1 ChR2 rats and hSyn D1ChR2 completely failed to self-stimulate laser illumination in this task, despite all previously being strongly and equally attracted to the laser-paired shock rod. **d** In both hSyn ChR2 and D1 ChR2 groups, some individuals self-stimulate laser at least at moderate >10 illuminations per session, but other individuals consistently do not self-stimulate. **e** A2 ChR2 rats ($N = 4$) failed to self-stimulate laser, and indicated a possible laser avoidance compared to the inactive spout/porthole. **f**, **g** Only a few CRF ChR2 ($N = 9$) individuals and control eYFP individuals ($N = 8$) self-stimulated laser illuminations, but this failed to reach significance for the groups. All data represent means and SEM. *$p < 0.05$, **$p < 0.01$.

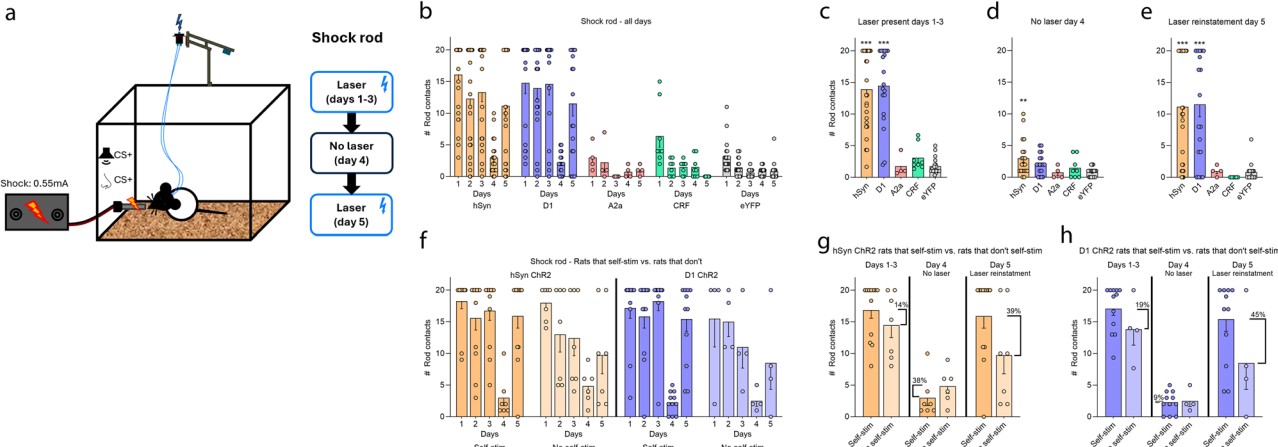

**Fig. 2 | CeA hSyn or D1 ChR2 creates shock rod attraction. a** Schematic diagram of shock rod chamber. Laser illuminations were paired whenever a rat was within ≤2 cm distance from shock rod. An auditory Pavlovian CS+ was also paired with laser activations, and a contextual odor CS+ was placed under the shock rod. **b** CeA hSyn ChR2 (N = 26) and D1 ChR2 (N = 19) rats showed high levels of approach and contacts with the laser-paired shock rod across Days 1–3, but A2a(D2) (N = 4) rats and CRF ChR2 (N = 8) rats did not. On Day 4 laser was discontinued (laser extinction trial). On Day 5 laser illuminations were resumed. **c** Average number of shock rod contacts on laser-paired days 1–3. **d** Contacts with shock rod nearly disappeared when laser was discontinued on day 4. **e** Contacts with the shock rod increased again when laser pairing was reinstated on day 5. **f** Both hSyn ChR2 (N = 12) and D1 ChR2 (N = 12) individuals that self-stimulated laser in the nose-poke/spout-touch task and individuals that failed to self-stimulate (hSyn ChR2 N = 7, D1 ChR2 N = 4) show robust shock rod attraction across days 1–3. Attraction disappeared on day 4 when laser was turned off (laser extinction), and reinstated on day 5 when laser illumination resumed. **g, h** For both hSyn ChR2 and D1 ChR2 groups, individuals that self-stimulate show modestly higher attraction to the shock rod on laser-paired days, although the difference was not statistically significant with these Ns. Data represent means and SEM. \*\*$p < 0.01$, \*\*\*$p < 0.001$.

self-stimulate hSyn ChR2 in the CeA may have added marginal extra attraction to the shock rod, rats that fail to self-stimulate still showed nearly as robust shock rod attraction as those that do, similar to previously reported findings[12].

**D1 ChR2 rats: attraction to shock rod.** On day 1, D1 ChR2 rats made a total of 14.79 ± 1.7 shock rod contacts. On Day 2, D1 ChR2 rats received 13.95 ± 1.69 shocks, and on Day 3 another 14.63 ± 1.74 shocks (Fig. 2b). Comparing shock rod attraction in D1 ChR2 individuals that did self-stimulate in the nose-poke/spout-touch task versus those that failed to self-stimulate, individuals that self-stimulated made 19% more shock rod contacts (17.08 ± 1.05) than those that failed to self-stimulate (13.83 ± 2.06). This difference was not statistically significant (independent-samples $t$-test, $t_{14} = 1.4$, $p = 0.18$), and there was no statistical correlation between laser self-stimulation and the strength of shock rod attraction (i.e., number of shocks received) (Pearson's correlation, $r_{15} = 0.43$, $p = 0.09$) (Fig. 2h). Thus, a propensity to self-stimulate may add a marginal extra D1 ChR2 attraction to shock rod, but even D1 ChR2 individuals that do not self-stimulate are robustly attracted to the laser-paired shock rod (See Supplementary Note 2 for further analyses).

**Comparison of D1 ChR2 and hSyn ChR2 attraction.** Overall, both hSyn ChR2 (13.91 ± 1.21) ($p < 0.001$) and D1 ChR2 rats (14.46 ± 1.36) ($p < 0.001$) received >700% more shocks across the three days than eYFP control rats (1.75 ± 0.27) (one-way ANOVA, $F_{4,72} = 29.77$, $p < 0.001$) (Fig. 2c). The hSyn ChR2 and D1 ChR2 rats did not differ significantly from each other in the total number of shocks received ($p = 1$), however, they did differ in detail of behavior patterns used to touch the shock rod with hSyn ChR2 rats making more brief paw/snout touches and D1 ChR2 rats making more longer duration chew contacts (Supplementary Fig. 1). Between receiving shocks, both D1 ChR2 rats and hSyn ChR2 rats also spent 800–1100% more time in close <2 cm proximity to the shock rod than eYFP control rats, amounting to 33–44% of the session (one-way ANOVA, $F_{4,46} = 18.15$, $p < 0.001$; hSyn ChR2 $p < 0.001$, D1 ChR2 $p < 0.001$) (Supplementary Fig. 2a). Rats that self-stimulated laser in the nose-poke/spout-touch task (hSyn ChR2 42%; D1 ChR2 44%) did not differ from rats that failed to self-stimulate in time spent near the shock rod (hSyn ChR2 37%, independent-samples $t$-test, $t_{10} = 0.79$, $p = 0.45$; D1 ChR2 42%; $t_{13} = 0.25$, $p = 0.81$) (Supplementary Fig. 2b).

**D1 ChR2 and hSyn ChR2 rats fail to show defensive burying to shock rod.** Neither hSyn ChR2 rats nor D1 ChR2 rats displayed much defensive burying behavior, despite receiving many shocks. For both, defensive burying was low on Day 1 (hSyn ChR2 cumulative duration = 13.41 ± 5.5 s; D1 ChR2 = 13.3 ± 3.96 s), amounting to only 1/10th that of eYFP control rats on Day 1 (128.85 ± 41.02 s; one-way ANOVA, $F_{4,47} = 5.52$, $p = 0.001$; hSyn $p = 0.012$; D1: $p = 0.016$) (Supplementary Fig. 2c). On Days 2 and 3, D1 ChR2 and hSyn ChR2 rats continued to emit relatively low levels of defensive burying (Day 2 hSyn = 16.17 ± 14.06 s; D1 = 16.2 ± 6.08 s; Day 3 hSyn = 25.18 ± 22.51 s; D1 = 13.85 ± 8.06 s). There was no significant difference in burying between individuals that self-stimulated laser versus those that did not, for either hSyn ChR2 (independent samples $t$-test, $t_5 = 1.29$, $p = 0.25$) or D1 ChR2 groups (independent-samples $t$-test, $t_{3.38} = 0.71$, $p = 0.53$) (Supplementary Fig. 2d).

**CRF ChR2 and A2(D2) ChR2 rats fail to show attraction to shock rod.** Transgenic CRF ChR2 rats and A2(D2) ChR2 rats showed no detectable attraction to their laser-paired shock rod. Instead, both CRF ChR2 rats and A2(D2) ChR2 rats avoided the shock rod similarly to control eYFP rats. Similar to eYFP controls, A2 ChR2 rats spent only 8.96% ±4.48 of the session in close <2 cm proximity to shock rod; much lower proximity time than either D1 ChR2 or hSyn ChR2 rats (D1 $p < 0.001$; hSyn $p = 0.031$; CRF $p = 1$; eYFP $p = 1$). Similarly, CRF ChR2 rats spent only 4.16% ±0.8 of the session in close proximity, much like eYFP control rats and less than either D1 ChR2 or hSyn ChR2 rats (D1 $p < 0.001$; hSyn $p = 0.002$; eYFP $p = 1$).

**CRF ChR2 and A2(D2) ChR2 rats show robust defensive behavior.** Both CRF ChR2 rats and A2(D2) ChR2 rats displayed defensive burying on Day 1, not significantly different from eYFP levels (CRF = 44.06 ± 31.97 s; A2(D2) = 163.36 ± 112.66 s; $F_{4,47} = 5.52$, $p = 0.001$; eYFP vs A2(D2):, $p = 1$; eYFP vs CRF: $p = 0.74$). CRF and A2(D2) ChR2 defensive burying became lower on Day 2 (CRF = 3.3 ± 2.37 s; A2(D2) = 27.56 ±

**Fig. 3 | CeA hSyn and D1 ChR2 rats are motivated to overcome an occluding barrier to reach the shock rod. a** Diagram of an opaque protective barrier placed between a rat and the shock rod. **b** CeA hSyn ($N = 17$) and D1 ChR2 ($N = 13$) rats repeatedly climbed over the barrier in order to reach and touch the shock rod. **c** Separately shows barrier crosses for hSyn ChR2 ($N = 11$) and D1 ChR2 ($N = 9$) individuals that self-stimulated laser in the nose-poke/spout-touch task versus individuals that failed to self-stimulate (hSyn ChR2 $N = 6$, D1 ChR2 $N = 4$). Data represent means and SEM.
*$p < 0.05$, **$p < 0.01$.

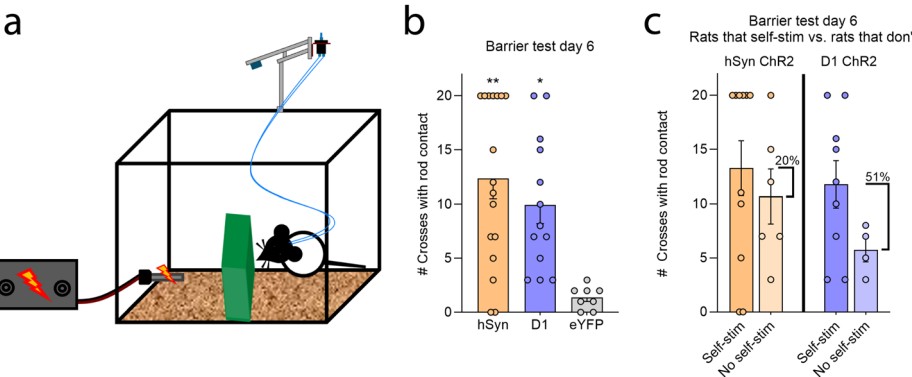

20.55 s), and Day 3 (CRF = 22.52 ± 17.21 s; A2(D2) = 2.24 ± 2.24 s) as these rats switched to avoiding the shock rod, staying mostly at the opposite end of the chamber, similarly to eYFP control rats.

In summary for Days 1–3, A2(D2) ChR2 rats and CRF ChR2 rats failed to show maladaptive attraction, and instead emitted defensive burying and then avoided the shock rod, similarly to eYFP control rats. By contrast, both hSyn ChR2 rats and D1 ChR2 rats were strongly attracted to the shock rod, did not show much antipredator behavior, and repeatedly touched the shock rod and received many shocks, both groups showing significantly stronger maladaptive attraction than eYFP control rats (hSyn $p < 0.001$; D1 $p < 0.001$), CRF ChR2 rats (hSyn $p < 0.001$; D1 $p < 0.001$), or A2(D2) ChR2 rats (hSyn $p < 0.001$; D1 $p < 0.001$) (one-way ANOVA, $F_{4,72} = 29.77$, $p < 0.001$) (See Supplementary video 1 for examples of shock rod attraction).

**Laser extinction day (Day 4).** On Day 4 laser was discontinued but the shock rod still gave shocks. Without laser stimulations of CeA, hSyn ChR2 rats almost immediately reduced their contacts to roughly 22% of their previous averaged level on Days 1–3 (3 ± 0.63 shocks; paired-samples $t$-test, $t_{19} = 8.14$, $p < 0.001$). However, despite having dropped, hSyn ChR2 rats still made more contacts than eYFP control rats, perhaps suggesting slight learned persistence ($p = 0.005$) (one-way ANOVA, $F_{4,64} = 3.86$, $p = 0.007$) (Fig. 2d). Comparing hSyn ChR2 individuals that self-stimulated laser in the nose-poke/spout-touch task to individuals that failed to self-stimulate, hSyn ChR2 rats that did self-stimulate did not make more shock rod contacts on the laser extinction day (3 ± 1.23) than hSyn ChR2 rats that failed to self-stimulate (4.83 ± 1.14; independent-samples $t$-test, $t_{11} = 1.08$, $p = 0.3$) (Fig. 2g).

D1 ChR2 rats dramatically reduced their shock rod attraction to only <15% their previous Days 1-3 level (2.11 ± 0.38; paired-samples $t$-test, $t_{17} = 8.96$, $p < 0.001$), and no longer differed from eYFP control rats, indicating a nearly complete dependence on activated brain state ($p = 0.37$) (Fig. 2d). D1 ChR2 rats that did self-stimulate in the nose-poke/spout-touch task did not make more shock rod contacts (2.27 ± 0.55) than D1 ChR2 rats that failed to self-stimulate (2.5 ± 0.71; independent-samples $t$-test, $t_{13} = 0.22$, $p = 0.83$) (Fig. 2h). We conclude that shock rod attraction requires the continued pairing of optogenetic brain states with shock rod encounters in order to be maintained at high levels.

CRF ChR2 (1.5 ± 0.6 shocks) and A2(D2) ChR2 rats (0.75 ± 0.48 shocks) continued to avoid the shock rod on laser-extinction day 4.

**Laser reinstatement (Day 5).** On the laser reinstatement day (Day 5), laser pairings resumed, and the shock rod still delivered shocks. With laser reinstated, hSyn ChR2 rats rose again in maladaptive attraction back to their own earlier attraction levels seen on laser-paired Days 1–3, now receiving 11.15 ± 1.65 shocks. This was significantly higher hSyn attraction than their level on the previous laser extinction Day 4 (paired-samples $t$-test, $t_{19} = 4.25$, $p < 0.001$), and also higher than eYFP control

rats on current Day 5 ($p < 0.001$) (one-way ANOVA, $F_{4,67} = 10.28$ $p < 0.001$) (Fig. 2e). Comparing self-stimulators to non-self-stimulators, hSyn ChR2 individuals that did self-stimulate made numerically more shock rod contacts (15.92 ± 1.91) than hSyn ChR2 rats that failed to self-stimulate (9.71 ± 2.94), but the difference was not statistically significant (independent-samples $t$-test, $t_{17} = 1.85$, $p = 0.08$) (Fig. 2g).

D1 ChR2 rats similarly rose again in shock rod attraction on the laser reinstatement Day 5 back to their own previous level of attraction on laser-paired Days 1–3, now receiving 11.53 ± 1.96 shocks. This D1 attraction level was significantly higher than their own level on the previous laser extinction Day 4 (paired-samples $t$-test, $t_{17} = 4.75$, $p < 0.001$), and higher than eYFP control levels on current Day 5 ($p < 0.001$) (Fig. 2e). D1 ChR2 rats that did self-stimulate made numerically more shock rod contacts (15.42 ± 1.68) than D1 ChR2 rats that failed to self-stimulate (8.5 ± 3.42), but this difference was not significant (independent-samples $t$-test, $t_{14} = 1.7$, $p = 0.11$) (Fig. 2h).

By contrast to D1 ChR2 and hSyn ChR2 rats, eYFP control rats, CRF ChR2 rats, and A2(D2) ChR2 rats continued to avoid the shock rod and received few or no shocks on the laser reinstatement Day 5.

## Motivated rod attraction overcomes obstacle

To assess if D1 ChR2 and hSyn ChR2 rats were appetitively motivated sufficiently to overcome an obstacle to reach and touch the shock rod, we used a 'barrier test' that was novel to them (Fig. 3a). Both hSyn ChR2 rats and D1 ChR2 rats repeatedly climbed over the opaque shoebox-sized barrier to reach and touch the shock rod, and consequently received multiple shocks. The hSyn ChR2 rats made 12.35 ± 1.85 crossings per 20-min session that culminated in shocks (plus 0.76 ± 0.32 unshocked crossings per session). D1 ChR2 rats made 9.92 ± 1.72 crossings that culminated in shocks per session (plus 2.85 ± 0.77 unshocked crossings per session). By comparison, control eYFP rats made fewer crossings followed by shocks than either hSyn ChR2 or D1 ChR2 rats (1.38 ± 0.38; one-way ANOVA, $F_{2,35} = 8.35$, $p < 0.001$; hSyn ChR2 $p = 0.001$, D1 ChR2 $p = 0.014$) per session (plus 5.75 ± 1.69 unshocked crossings) (Fig. 3b).

We compared crossings by individuals that self-stimulated laser in the nose-poke/spout-touch versus those that did not self-stimulate: hSyn ChR2 individuals that self-stimulated made slightly more shocked barrier crosses (13.27 ± 2.54) than individuals that failed to self-stimulate (10.67 ± 2.54), but this difference was not statistically significant (independent-samples $t$-test, $t_{15} = 0.66$, $p = 0.52$). Suggesting a potentially larger role of self-stimulation in D1 ChR2 rats, individuals that did self-stimulate made 51% more barrier crosses that culminated in shock (11.78 ± 2.14) than individuals that failed to self-stimulate (5.75 ± 1.12), though again the difference was not statistically significant (independent-samples $t$-test, $t_{11} = 1.76$, $p = 0.11$). Overall, even hSyn ChR2 and D1 ChR2 rats that failed to self-stimulate were motivated to repeatedly cross a barrier to interact with the shock rod at robust levels (Fig. 3c; see Supplementary Fig. 3 for further analyses).

**Fig. 4 | CeA hSyn and D1 ChR2 rats do not show attraction to a dummy rod. a** Diagram depicts the dummy rod situation, where an unelectrified dummy rod was similar in appearance to the shock rod, and delivered laser illuminations, but did not inflict electric shock. **b** hSyn ChR2 ($N = 7$) and D1 ChR2 ($N = 12$) all failed to show significant attraction to the dummy rod above eYFP control levels ($N = 4$), and attraction to the dummy rod was not higher on the day when it delivered laser illumination than the day when it delivered no laser. **c** For both hSyn ChR2 ($N = 3$) and D1 ChR2 ($N = 6$) groups, individuals that self-stimulate laser did not differ from those that failed to self-stimulate (hSyn ChR2 $N = 4$, D1 ChR2 $N = 3$) in attraction to the dummy rod when it delivered laser. Data represent means and SEM.

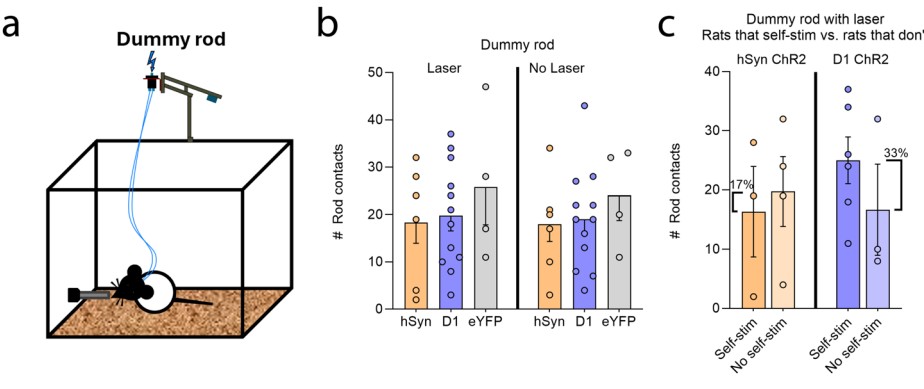

## Dummy rod attraction

Dummy rod encounters never gave shocks, and was tested both with and without laser pairings (Fig. 4a). Neither hSyn ChR2 nor D1 ChR2 rats made more touches on the dummy rod than eYFP control rats on the test day when each touch delivered CeA laser (repeated-measures ANOVA, $F_{2,20} = 0.14$, $p = 0.712$; hSyn = 18.29 ± 4.32; D1 = 19.75 ± 3.2; eYFP = 25.75 ± 7.91). Further, neither hSyn ChR2, D1 ChR2, nor eYFP control rats touched the dummy rod more on their Laser-delivering day than on the No Laser day (hSyn = 18 ± 3.68, D1 = 19 ± 3.11, eYFP = 24 ± 5.24). (Fig. 4b; See Supplementary Fig. 4 for further analyses).

Comparing individuals that did self-stimulate laser in the nose-poke/spout-touch task versus individuals that failed to self-stimulate, hSyn ChR2 rats that did self-stimulate made slightly fewer dummy rod contacts on the laser day (16.33 ± 7.62) than hSyn ChR2 rats that failed to self-stimulate (19.75 ± 5.89), but the difference was not significant (independent-samples $t$-test, $t_5 = 0.36$, $p = 0.73$). D1 ChR2 rats that did self-stimulate made more dummy rod contacts (25 ± 3.97) than D1 ChR2 rats that failed to self-stimulate on the laser day (16.67 ± 5.96), but again this difference was not significant (independent-samples $t$-test, $t_7 = 0.31$, $p = 0.77$) (Fig. 4c).

Overall, the relatively low attraction of D1 ChR2 or hSyn ChR2 rats to the dummy rod, compared to shock rod, supports the counter-intuitive conclusion that UCS shocks from the shock rod positively contribute to maladaptive attraction (when combined with paired CeA ChR2 stimulations), rather than deter attraction as a price to be reluctantly paid by rats seeking to self-stimulate CeA laser[12].

## Conditioned reinforcement & preference tests: are shock cues sought out?

**Instrumental conditioned reinforcement test.** A hallmark of incentive salience is that it can make reward-associated Pavlovian cues become attractive and 'wanted'. In a conditioned reinforcement task, where rats could make nosepokes to earn brief auditory CSs that previously had been paired with shocks from shock rod encounters, hSyn rats made nearly 150% more nosepokes overall on the day they earned their shock-associated CS+ sound (13 ± 2.1 nosepokes) than on the day they earned the safe CS− sound associated with home cage (8.75 ± 1.32; paired-samples $t$-test, $t_7 = 2.31$, $p = 0.05$) (Fig. 5a, b). The higher number of hSyn ChR2 nosepokes on the CS+ day were distributed across both portholes ($t_7 = 0$, $p = 1$), suggesting rats were excited by CS+ presentations, but generalized across portholes and did not learn in the single trial to discriminate between the CS+ port (6.5 ± 1.13) and inactive port (6.5 ± 1.21). On the CS− day hSyn ChR2 made fewer nosepokes, and again equally distributed between the active porthole (5.13 ± 0.97) and inactive porthole (3.63 ± 0.56) (paired-samples $t$-test, $t_7 = 1.69$, $p = 0.13$).

D1 ChR2 rats similarly made over 200% more total nosepokes on the day they earned their shock-associated CS+ (34.67 ± 7.71) than on their safe CS− day (17 ± 4.16; paired-samples $t$-test, $t_5 = 2.64$, $p = 0.046$), again distributed across both portholes (Fig. 5c).

Thus, overall it appeared that for both hSyn ChR2 rats and D1 ChR2 rats, the shock-associated CS+ sound was more highly motivating than the CS− sound. Both groups of rats similarly generalized between portholes during the single 30-min session used on both days here, and did not successfully learn the instrumental discrimination (only a single instrumental session was given because it was an extinction trial that would have weakened subsequent CS-UCS associations). However, we note that in a previous CeA ChR2 study of conditioned reinforcement using an auditory CS+ for shock rod, hSyn ChR2 rats were both more highly motivated by CS+ than CS−, and also successfully discriminated between portholes, specifically making more nosepokes into the CS+ porthole than into any of the other three portholes[12].

Control eYFP rats did not differ in total nosepoke responses between the CS+ day (9.33 ± 1.26) and CS− day (13.33 ± 3.2; paired-samples $t$-test, $t_5 = 1.09$, $p = 0.33$).(Fig. 5d). Overall, on the CS+ day, D1 ChR2 rats made more total nosepoke responses than either hSyn ChR2 rats ($p = 0.006$) or eYFP rats ($p = 0.003$; one-way ANOVA, $F_{2,17} = 9.47$, $p = 0.002$) (Fig. 5e). Total nosepoke responses on the CS− day did not differ between any of the groups (one-way ANOVA, $F_{2,17} = 2.19$, $p = 0.14$) (Fig. 5f). A pilot test with naïve control rats, which had no previous exposure to the tone or white noise, indicated that rats had no pre-existing preference between the auditory stimuli used as CSs (paired-samples $t$-test, $t_7 = 0.3$, $p = 0.78$) (Fig. 5g; see Supplementary Note 3 for further analyses).

**Olfactory cue preference test: is a shock-associated contextual CS + odor sought out?**. In this separate conditioned reinforcement test, rats could seek out contextual CS odors in different chambers (Fig. 6a). The hSyn ChR2 rats preferred their CS+ odor associated with shock rod (550.13 sec ±50.89) over the home-associated CS− odor, spending 150% more time in the shock-associated CS+ scented chamber than in the safe CS− scented chamber (348.88 sec ±50.89; paired-samples $t$-test, $t_7 = 1.97$, $p = 0.09$) (Fig. 6b).

Similarly, D1 ChR2 rats also preferred their CS+ odor over the CS− odor, spending 150% more time in the shock-associated CS+ scented chamber (571.09 sec ±34.25) than in the safe-scented CS− chamber (328.91 s ±34.25; paired-samples $t$-test, $t_{10} = 3.54$, $p = 0.005$) (Fig. 6c).

Overall, these results indicate that both hSyn ChR2 rats and D1 ChR2 rats are willing to seek out a contextual odor CS+ previously associated with the shock rod: both preferred the CS+ over a safe CS− odor associated with home cage. That is, the shock-associated CS+ odor was sought out as if it were a reward cue, even if it initially had aversive sensory features (see Supplementary Fig. 5 for further analyses).

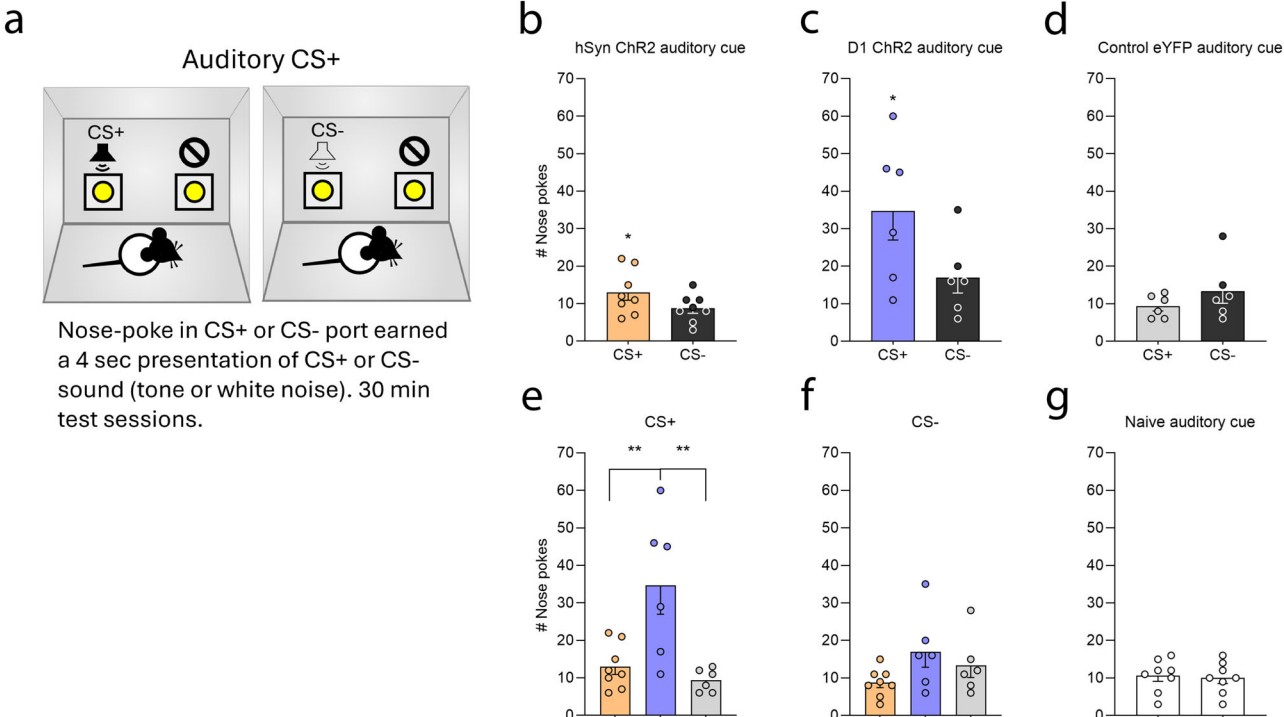

**Fig. 5 | CeA hSyn and D1 ChR2 rats seek auditory Pavlovian cue that was previously paired with shock rod encounters. a** Diagram depicts the instrumental conditioned reinforcement task, in which rats could nosepoke to earn brief presentations of the shock-associated auditory CS+ (tone or white noise) that was previously paired with shock rod contacts, or the safe CS− sound (whichever noise/tone was not the CS+) that had previously been heard in the home cage. On one day, nosepokes into a designated active porthole earned the shock-associated CS+, whereas nosepokes into the alternative inactive porthole earned nothing. On another day, nosepokes in the active porthole earned presentations of the safe CS− sound, while nosepokes in the inactive porthole earned nothing. **b** hSyn ChR2 (*N* = 8) rats made more nosepokes on the day they earned their shock-associated CS+ sound than on the day they earned their safe CS− sound. **c** D1 ChR2 (*N* = 6) rats similarly made twice as many nosepokes on the day they earned their shock-associated CS+ sound than on the day they earned their safe CS− sound. **d** Control eYFP rats (*N* = 6) made relatively few nosepokes for either CS, and did not differ between CS+ or CS− days. **e** D1 ChR2 rats responded more on the CS+ day than hSyn ChR2 and control eYFP rats. **f** hSyn ChR2, D1 ChR2, and control eYFP rats did not respond differently on the CS− day. **g** Naïve unoperated control rats (*N* = 8) did not have a preference between tone or white noise used as CSs. Data represent means and SEM. \**p* < 0.05, \*\**p* < 0.01.

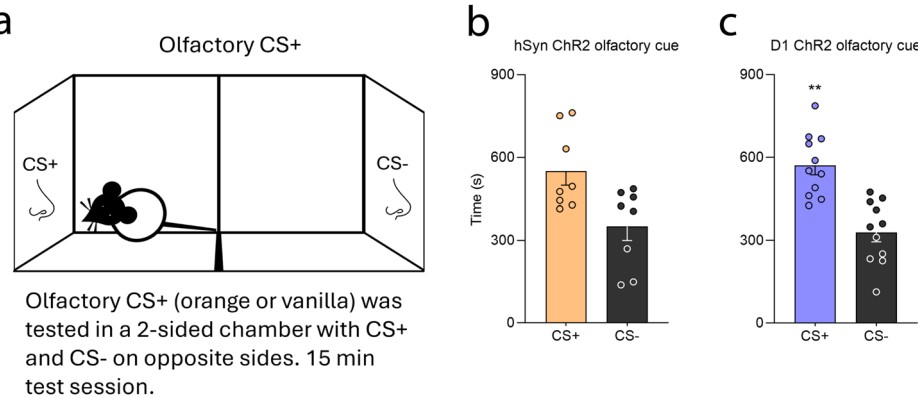

**Fig. 6 | CeA hSyn and D1 ChR2 rats seek olfactory contextual CS+ cue that was previously paired with shock rod. a** Diagram depicts the 2-chamber odor preference test, in which rats could choose to approach either a shock-associated contextual CS+ scent that had previously been placed under the shock rod in one chamber (vanilla or orange extract), or approach the alternative safe CS− scent that had previously been placed in the home cage in the other chamber (whichever was not CS+). **b** hSyn ChR2 rats (*N* = 8) spent more time in the shock-associated CS+ scented chamber than in the safe CS− scented chamber. **c** D1 ChR2 rats (*N* = 8) similarly spent more time in the shock-associated CS+ scented chamber than in the safe CS− scented chamber. Data represent means and SEM. \*\**p* < 0.01.

**Histological analysis of CeA virus and laser-induced local CeA Fos plumes**

Virus infections typically had a diameter of 1.6–2 mm in CeA, and sometimes also extended slightly out of CeA into globus pallidus, neostriatum, or basolateral amygdala (Fig. 7a, b). By comparison, local Fos plumes induced by laser ChR2 stimulation typically were smaller than virus infections and were centered around the optic fiber tip site within the virus zone, similar to previous studies (Warlow et al., 2020). Laser stimulation in hSyn ChR2 rats produced local CeA Fos plumes of radius 0.2–0.35 mm around the fiber tip (0.4–0.7 mm diameter), where Fos expression was >300% elevated over

eYFP baselines (independent-samples *t*-test, $t_{14}$ = 14.31, *p* < 0.001). Similarly, in D1 ChR2 rats, laser stimulation produced local Fos plumes of >300% elevation with a radius 0.2–0.35 mm (0.4–0.7 mm diameter) (independent-samples *t*-test, $t_{13}$ = 6.55, *p* < 0.001) (Fig. 8a, b; see Supplementary Fig. 6 for mRNA verification of Cre on D1 receptor-expressing neurons).

**Shock rod attraction recruits widespread neurobiological activation of mesocorticolimbic circuitry: distant Fos analysis**

Shock rod attraction in hSyn ChR2 rats also recruited significant Fos increases over eYFP control levels in several other mesocorticolimbic brain

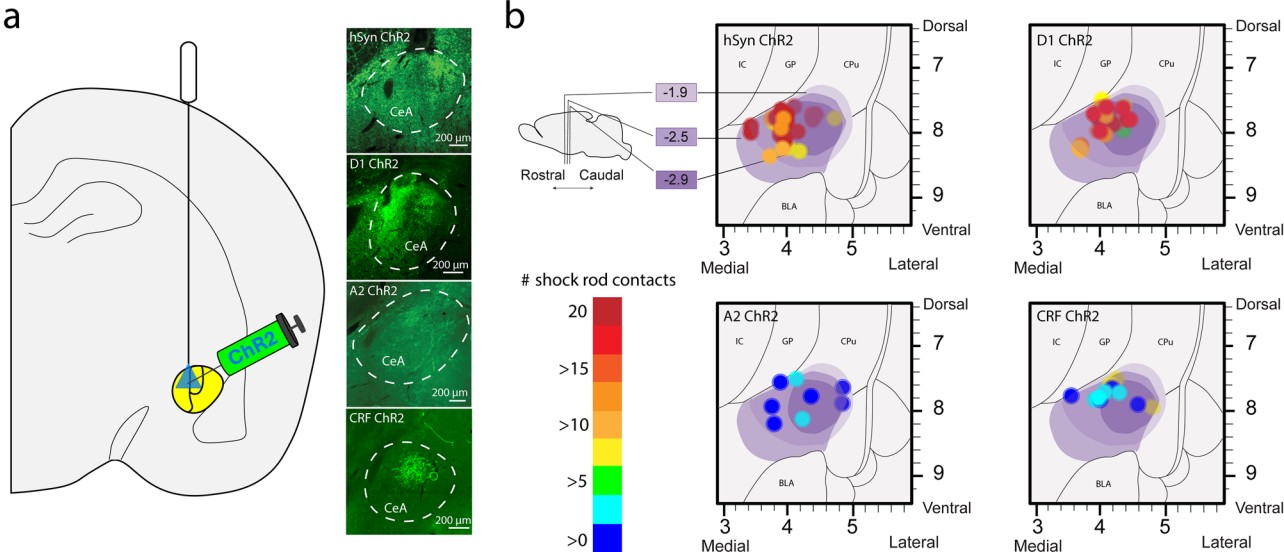

**Fig. 7 | AAV expression and Placements Verification. a** Schematic diagram and photomicrographs of CeA where AAV-hSyn-ChR2-eYFP, AAV-DIO-ChR2-eYFP, AAV-hSyn-eYFP, or AAV-DIO-eYFP was microinjected bilaterally staggered around coordinates A/P from Bregma in mm: −2.4, M/L: ±4.1, D/V: −7.8[35]. hSyn, D1 Cre, A2 Cre, and CRF Cre-targeted neurons expressing ChR2-eYFP are shown in green. **b** Schematic diagram showing surgical placements in hSyn, D1, A2, and CRF ChR2 rats with placement circles color-coded by number of shock rod contacts made by the individual rat corresponding to that site across shock rod test days 1–3 on a color gradient of red (20 contacts) to blue (0 contacts).

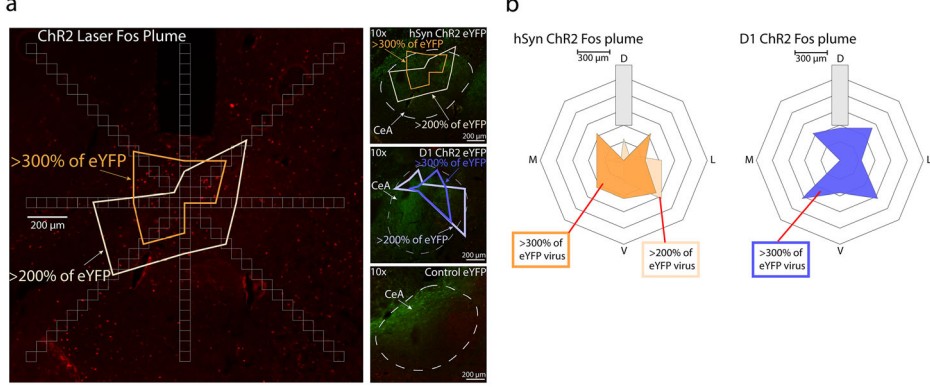

**Fig. 8 | Local CeA Fos plumes. a** Photomicrographs show laser-induced elevation of Fos expression as local CeA Fos plumes. **b** Diagram shows average size of Fos plumes, in which Fos expression was elevated >300% over eYFP baselines, radiating 0.2–0.35 mm around the fiber tip in hSyn and D1 ChR2 rats.

structures (Fig. 9a): >300% Fos elevation in the prelimbic subregion of the anterior cingulate cortex (PL) (one-way ANOVA, $F_{3,36} = 16.13$, $p < 0.001$; hSyn $p < 0.001$), >500% in dorsomedial striatum (DMS) (one-way ANOVA, $F_{3,36} = 15.79$, $p < 0.001$; hSyn $p < 0.001$), >500% in the rostral half of nucleus accumbens medial shell (NAcsh) (one-way ANOVA, $F_{3,36} = 10.68$, $p < 0.001$; hSyn $p = 0.01$), >300% in the caudal half of NAc medial shell (one-way ANOVA, $F_{3,36} = 17.05$, $p < 0.001$; hSyn $p = 0.001$), >200% in medial amygdala (MeA) (one-way ANOVA, $F_{3,36} = 7.54$, $p < 0.001$; hSyn $p = 0.002$), and >300% in the ventral tegmental area (VTA) (one-way ANOVA, $F_{3,36} = 18.94$, $p < 0.001$; hSyn $p < 0.001$) (Figs. 9b and 10).

Similarly in D1 ChR2 rats, shock rod attraction recruited >300% distant Fos increases in PL (D1 $p < 0.001$), >200% in infralimbic cortex (one-way ANOVA, $F_{3,36} = 4.15$, $p = 0.013$; D1 $p = 0.047$), >500% in the DMS (D1 $p = 0.004$), >500% in the rostral NAcsh (D1 $p < 0.001$), >400% in the caudal NAcsh (D1 $p < 0.001$), >300% in the rostral nucleus accumbens core (NAcc) (one-way ANOVA, $F_{3,36} = 4.7$, $p = 0.007$; D1 $p = 0.01$), >500% in the caudal NAcc (one-way ANOVA, $F_{3,36} = 6.25$, $p = 0.002$); (D1 $p = 0.001$), >400% in the lateral hypothalamus (LH) (one-way ANOVA, $F_{3,36} = 6.04$, $p = 0.002$; D1 $p = 0.005$), >300% in the MeA (D1 $p < 0.001$), and >200% in the VTA (D1 $p = 0.01$) (Figs. 9b and 10). Altogether, these patterns seem consistent with the hypothesis that activation of mesocorticolimbic circuitry of incentive motivation is the potential mechanism underlying maladaptive attraction to the shock rod[12].

## Discussion

Our results demonstrate that selective ChR2 stimulation of CeA neurons with D1 dopamine receptors is sufficient to induce maladaptive attraction to a noxious shock rod in rats, when paired with shock rod encounters, comparable in intensity to that induced by general hSyn stimulation of all CeA neuronal subtypes. By contrast, selective stimulations of CeA neurons with either D2 receptors and A2 adenosine receptors, or that use CRF as a neurotransmitter, failed to induce shock rod attraction.

Our findings confirm a previous report that hSyn CeA ChR2 stimulation, when associatively paired with shock rod encounters, can generate maladaptive 'wanting what hurts'[12]. Our current results further pinpoint CeA D1 neurons as a particular subpopulation sufficient by itself to induce appetitive shock rod attraction. Whether any other single subpopulation also might be sufficient too, or whether multiple other subpopulations that are individually insufficient might be sufficient if co-activated together without CeA D1 neuronal co-activation, remain open questions.

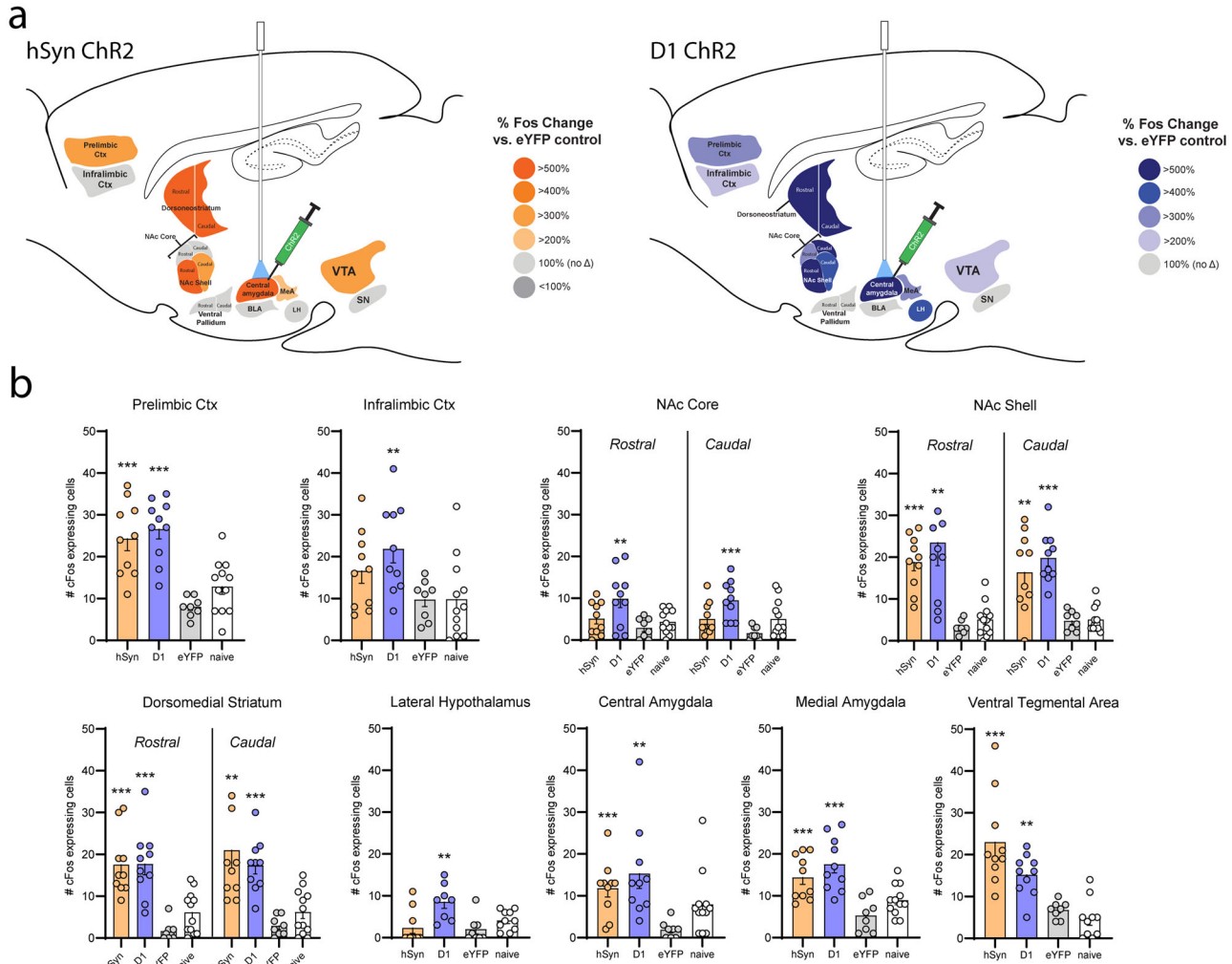

**Fig. 9 | Distant Fos elevations: recruitment of mesocorticolimbic circuitry during maladaptive attraction to shock rod. a** Diagram shows distant brain structures in which Fos elevations were recruited in hSyn ChR2 rats ($N = 10$; orange) or D1 ChR2 rats ($N = 10$; blue) during shock rod attraction. **b** Shows quantitative levels of Fos expression in D1 ChR2 and hSyn ChR2 rats immediately after shock rod attraction, as well as in eYFP control rats and naïve unoperated control rats. Structures include: prelimbic cortex (PL), infralimbic cortex (IL), medial shell of nucleus accumbens (NAcsh), core of nucleus accumbens (NAcc), dorsomedial neostriatum (DMS), medial amygdala (MeA), lateral hypothalamus (LH), and ventral tegmentum (VTA). Data represent means and SEM. $**p < 0.01$, $***p < 0.001$.

Maladaptive shock rod attraction induced by CeA ChR2 stimulation also recruited activation of distant mesocorticolimbic circuitry related to incentive salience, as indicated by increased Fos expression in limbic structures, in both D1 ChR2 and hSyn ChR2 rats. Elevation in Fos expression was observed in the ventral tegmentum, shell and core of nucleus accumbens, dorsomedial neostriatum, anterior cingulate cortex (prelimbic and infralimbic regions), and related limbic structures. Such patterns resemble mesocorticolimbic activation elicited during CeA ChR2 pursuit of conventional sucrose or cocaine rewards[12,17,18]. Recruitment of activation in mesocorticolimbic structures underlying incentive motivation may provide neural mechanisms for generating maladaptive attraction.

### Incentive motivation underlies shock rod attraction

Several observations indicated that shock rod attraction was mediated by an appetitive motivational process, such as incentive salience, rather than by defensive or aggressive motivation elicited by shocks. For example, both D1 ChR2 and hSyn ChR2 rats repeatedly climbed over a protective barrier to actively seek out the shock rod, rather than remain on the safe side of the barrier[12]. Further, D1 ChR2 and hSyn ChR2 rats sought out olfactory and auditory Pavlovian CS+ cues that were associated with shocks from the shock rod, rather than avoid those shock-associated cues. Normally, only reward cues are 'wanted' in such fashion, but CeA D1/hSyn ChR2 pairing

appears to make cues for the noxious shock rod become appetitively 'wanted' as if they were reward cues.

Shock rod attraction itself may be a form of appetitive sign-tracking to the rod that has become a CS+ cue for paired laser/shock UCS, making it a potent 'motivational magnet'. If so, it would be more accurate to conclude that D1 ChR2 and hSyn ChR2 rats 'wanted' the shock rod as an attractive CS+ object, similar to 'wanting' reward CS+s, rather than that they 'wanted' to receive electric shocks as UCSs (which may have remained noxious).

### Do shocks remain painful during CeA ChR2 stimulation?

The CeA can modulate pain perception[19], so it is conceivable that CeA ChR2 stimulation reduced the perceived pain of electric shock. However, several considerations suggest shock rod contacts still remained noxious during CeA ChR2 stimulations. First, aversive flinches as quick jerks of paw or mouth away from the shock rod were still often evoked from both D1 ChR2 and hSyn ChR2 rats during CeA ChR2 laser stimulations. Further, when CeA laser pairing was discontinued in the shock rod chamber on Day 4, both hSyn and D1 rats almost immediately began to avoid the shock rod as if they already had learned it was aversive and were ready to respond defensively as soon as the brain state induced by CeA ChR2 pairings was removed. In addition, the dummy rod, which delivered CeA laser illumination but no pain at all, was less able than the shock rod to promote

**Fig. 10 | Elevated Fos expression.** Photo-micrographs showing examples of elevated Fos expression in various mesocorticolimbic structures of hSyn ChR2 and D1 ChR2 rats.

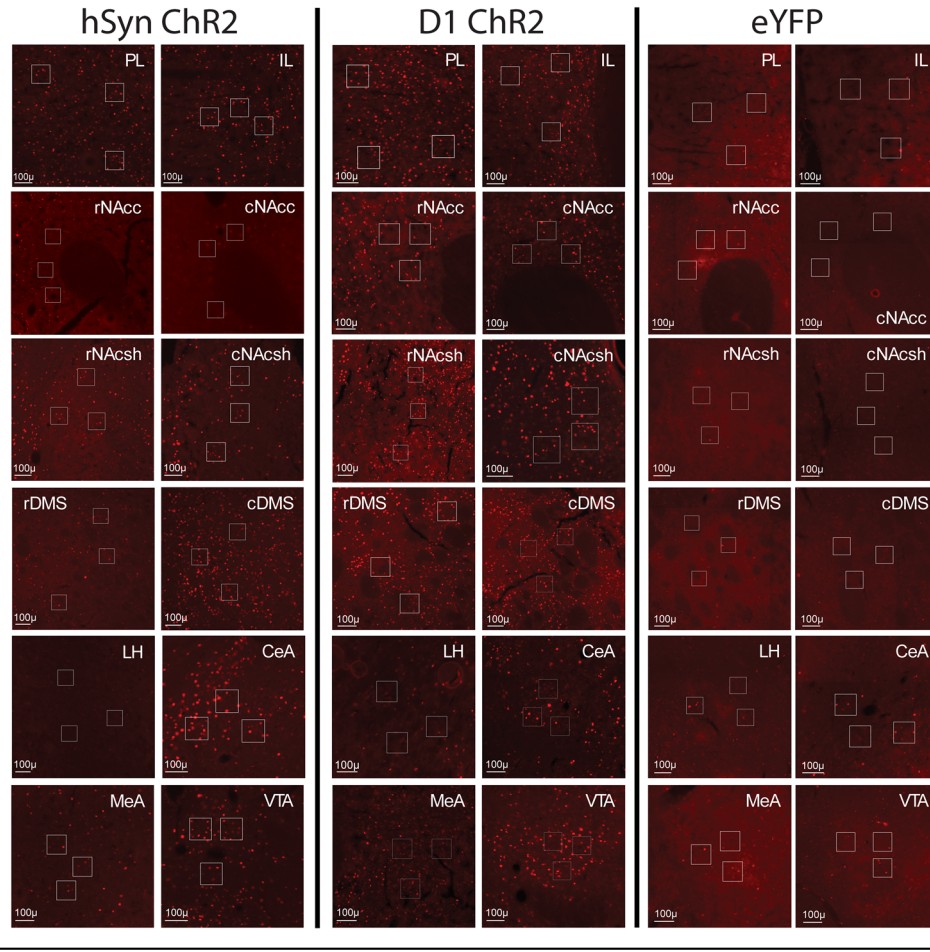

attraction above eYFP baselines in D1 ChR2 or hSyn ChR2 rats. Finally, in a previous study, CeA hSyn ChR2 laser stimulation was reported to increase negatively-valenced defensive conditioned reactions of freezing and avoidance when paired with a Pavlovian fear conditioning paradigm[12]. Increase in defensive reaction is the opposite outcome from what would be expected if CeA ChR2 stimulation simply reduced pain. We conclude that CeA-induced analgesia by itself is not likely to be the primary mechanism underlying shock rod attraction.

## Oral consummatory reactions

Another signature feature of a CS+ attributed with incentive salience is that it often elicits oral consummatory actions from rats, such as chewing a metal cue associated with sucrose rewards or i.v. cocaine rewards[8,11,17,18]. Here, 73% of hSyn ChR2 rats and 100% of D1 ChR2 rats nibbled, bit or chewed the shock rod at least once, despite incurring shocks on mouth or teeth. Consummatory chewing was especially prominent in CeA D1 rats, which may be related to reports by others that optogenetic inhibition of CeA D1 neurons oppositely reduces consummatory eating and drinking[7,20]. Similarly, CeA somatostatin neurons, which may also express D1 receptors[7], show increased calcium activity during eating[21], although inhibition of CeA somatostatin neurons paradoxically is also reported to increase food seeking[22].

## Laser self-stimulation does not fully explain maladaptive attraction

Did D1 ChR2 and hSyn ChR2 rats seek the shock rod primarily to gain CeA laser self-stimulation? The answer appears to be essentially no, although a self-stimulation motive may have added additional attraction for some individuals. The CeA does support optogenetic laser self-stimulation in several studies when rats are assessed as an entire group[7,12,23–25], though not in all studies[17,18]. However, there is striking within-group variation. For

example, some hSyn ChR2 and D1 ChR2 individuals self-stimulated intensely here in the nose-poke/spout-touch task (i.e., 50 to over 1000 illuminations per session) or at least moderately (10–49 illuminations), but nearly 40% of hSyn ChR2 individuals and 20% of D1 ChR2 individuals completely failed to self-stimulate CeA laser on any of the three test days. We found that hSyn ChR2 and D1 ChR2 individuals that self-stimulated did tend to maintain moderately stronger shock rod attraction across Days 2–3, and stronger reinstatement on Day 5 when laser was returned after laser extinction. But a number of hSyn ChR2 and D1 ChR2 rats that failed to self-stimulate showed as robust shock rod attraction as rats that did self-stimulate. Several rats that failed to self-stimulate had to be removed early from shock rod sessions because they reached the 20 shock maximum. Also, we note that hSyn ChR2 and D1 ChR2 rats could have self-stimulated laser and yet not receive shocks, by merely staying within 2 cm of the shock rod but not touching it, since close proximity alone was sufficient to trigger laser illumination. Yet, no hSyn ChR2 or D1 ChR2 rat exploited that opportunity. Overall, laser self-stimulation does not appear to be a complete or even chief explanation of CeA ChR2-induced shock rod attraction, and perhaps not at all for individuals that failed to self-stimulate in situations without shock.

## Potential explanation of motivational paradox?

Shock rod attraction in rats that otherwise fail to self-stimulate laser suggests that UCS shocks from the shock rod actually facilitated rather than deterred attraction (when coupled with CeA D1 ChR2 or hSyn ChR2 stimulation)[12]. That presents a paradox to conventional reinforcement theories. Warlow et al. suggested a potential explanation for this paradox: CeA optogenetic stimulation assigns intense incentive salience to the paired shock rod or to a reward CS+ when two simultaneous neural events occur together: 1) endogenous brain limbic activation triggered by an affective UCS that is either pleasant (e.g., cocaine or sucrose) or moderately unpleasant (e.g.,

shock rod), combined with 2) optogenetic neuronal excitation of D1 neurons or additional hSyn-targeted neurons in CeA[12,26]. It may seem surprising that moderately noxious electric shocks from the shock rod can serve to support CeA ChR2-induced attraction similarly to cocaine or sucrose rewards. This may possibly reflect a degree of neural overlap in limbic circuitry patterns activated by cocaine, sucrose, and even shock rod UCS encounters, allowing all to support incentive salience attribution. In the case of the shock rod, this overlap in incentive salience circuitry is sufficient to reverse the shock rod's motivational valence from defensive threat to being appetitively 'wanted'[26,27]. But in the absence of such affective UCSs, the incentive motivation to seek CeA ChR2 laser self-stimulation by itself remains relatively weak by comparison, or even absent for some rats.

### Precedents for 'wanting what hurts'?

Maladaptive 'wanting what hurts' may seem a surprising phenomenon, but there are several precedents in the literature. Perhaps the closest parallel is a report that rat pups younger than 10 days old develop a conditioned preference for an odor CS+ associated with electric shock[28]. It is not until day 21 that pups invariably form a conditioned avoidance for a shock-paired odor. Speculatively, an early-life capacity to reverse the motivational valence of an aversive UCS into a conditioned incentive attraction to its CS+ may presage an adult overlap in CeA-related and mesocorticolimbic brain circuitry that allows a shock rod and its cues to become positively 'wanted'. Another relevant valence-reversal may be the maladaptive attraction of adult rodents to a normally-avoided predator odor of cat urine, induced after brain infections by the protozoan parasite toxoplasma gondii, which can create cysts in amygdala and other limbic structures[29–32].

Yet even ordinary human adults may sometimes seek out electric shock if sufficiently bored[33]. For example, Wilson et al. described adult volunteers who said they would be willing to pay money to avoid shocks after receiving a sample shock[33], yet pressed the shock-producing button when subsequently left alone with nothing to do for 15 min. Several individuals pressed once or twice, and so received one or two shocks. But one participant self-administered 190 shocks, conceivably reflecting a more positively-valenced form of 'wanting what hurts' related similar to our results[33]. More generally, human thrill seekers who pursue experiences that others find aversive may well involve lesser but related reversals of motivational valence (e.g., roller coasters, parachute jumping, horror films, etc.).

### Implications for clinical disorders

We suggest maladaptive CeA ChR2 'wanting what hurts' may be a laboratory prototype of addictive-like motivation, as addiction is viewed by the incentive sensitization theory of addiction[15,34]. That is, as intense and narrowly-focused incentive salience or 'wanting' that can become independent of hedonic 'liking' for the same target. Of course, no addicted human has ever received optogenetic brain stimulations. But CeA ChR2 pairing in rats might powerfully and quickly in minutes to days induce changes in mesocorticolimbic circuitry of incentive salience similar to those that occur endogenously and gradually over longer periods in vulnerable humans. If so, our finding that CeA D1 neurons play a special role in ChR2-induced 'wanting what hurts' may also imply a special capacity for CeA D1 neuronal signaling to recruit mesocorticolimbic circuitry underlying sensitized incentive salience that creates human addictions (See Supplementary Note 4 for study limitations).

## Methods
### Animals

Male and female Sprague-Dawley and Long Evans wildtype rats ($N = 35$; male = 22, female = 13), transgenic D1-Cre Long Evans rats ($N = 24$; male = 8, female = 16), transgenic A2-Cre Long Evans rats ($N = 6$; male = 4, female = 2), and transgenic *Crh*-Cre Wistar rats ($N = 10$; male = 7, female = 3), were used weighing between 230 g and 400 g at surgery. Male and female rats were housed in separate rooms maintained at ~21 °C on a reverse 12-h light/dark cycle. All rats had ad libitum access to food (Purina

Lab Chow) and water in their home cages throughout the experiment. Prior to beginning behavioral testing, rats were handled for at least 2 days for 10 min each day. All procedures were approved by the University of Michigan's Animal Care and Use Committee. We have complied with all relevant ethical regulations for animal use.

### Optogenetic virus infusion and optic fiber implant

Rats were anesthetized with isoflurane gas (induction: 5%, maintenance: 2–3%), and received atropine (0.04 mg/kg; intraperitoneal, Henry Schein) and carprofen (5 mg/kg, subcutaneous, Henry Schein) prior to stereotaxic surgery. Adeno-associated virus (AAV) containing a channelrhodopsin 2 (ChR2) gene, either driven by the human synapsin promoter (hSyn) (AAV5-hSyn-ChR2-eYFP; 0.75 µl) for general neuronal targeting in wild-type rats, or double floxed inverted (AAV5-DIO-ChR2-eYFP; 1 µl) for Cre-mediated targeting was microinjected bilaterally into the CeA (A/P from Bregma in mm: −2.4, M/L: ±4.1, D/V: −7.8) with mouth bar set to −3.3; 0.1 µl/min). Sites were slightly staggered across rats to be distributed throughout the CeA, but were bilaterally identical within each rat. Control rats received AAV without ChR2 (AAV5-hSyn-eYFP, AAV5-DIO-eYFP) with identical microinjection procedures and CeA sites. Bilateral 200 µm core diameter optic fibers were implanted 0.3 mm directly above the intended CeA site. Fibers were secured in place with skull screws and dental acrylic cement. Cefazolin sodium (60 mg/kg, subcutaneous, Henry Schein) was administered to prevent infection. Carprofen (5 mg/kg, subcutaneous, Henry Schein) was given as post-surgical analgesia once per day for two days after surgery.

### Laser-paired shock rod tests: Days 1–3

The 'shock rod' was a metal rod (9 cm long and 1 cm diameter) wrapped in electrified metal wire that protruded from a wall of a Plexiglass chamber (38 cm length × 38 cm width × 48 cm height). The floor was covered with 2 cm cob bedding to enable defensive burying, an anti-predator behavior (Bed'O'Cobs, Andersons Inc., Maumee).

Contact with the shock rod was never forced, so always was completely voluntary. As the shock rod occupied approximately only 2% of the chamber's floor space, it could be easily avoided. Rats received paired optogenetic laser stimulations (473 nm, either 10 mW at 40 Hz or 3 mW at 25 Hz for different rats) whenever they voluntarily came within ~2 cm proximity to the shock rod, and laser was turned off as soon as the rat moved outside the 2 cm proximity zone. Varying the laser intensity and frequency helped assess whether induction of shock-rod attraction was a robust CeA phenomenon that could be reliably elicited by more than one stimulation configuration or instead an idiosyncratic phenomenon limited to only one particular stimulation configuration. When voluntarily touched, the shock rod delivered a mild electric shock (~0.2–0.5 mA measured using in-house ammeter). Variations could occur depending on duration and type of voluntary contact because while the shock generator was always set to 0.5 mA, and contacts with paw skin or mouth consistently produced 0.5 mA in measured amplitude, other contacts that were partly insulated by fur or nails sometimes produced lower amplitudes between 0.2 mA and 0.5 mA.

Initial shock rod testing was carried out in 3 daily 20-min laser-paired sessions. A maximum ceiling of 20 shocks per session was imposed to prevent unnecessary suffering for rats that were highly attracted to the shock rod. Each session continued for either 20 min or until the rat reached its maximum ceiling of 20 allowed shocked contacts, whichever came first, upon which the rat was gently removed.

### Laser extinction: Day 4

The fourth day was a laser extinction session, in which no laser was illuminated but the shock rod still delivered shocks. This was done to assess whether maladaptive attraction was a learned and enduring response that persisted once established by the 3 laser-paired sessions, or whether attraction instead required concomitant CeA ChR2 stimulation in the same session to induce an activated brain state.

## Laser reinstatement: Day 5

On the fifth day, laser activation was reinstated, and procedures were identical to the first 3 sessions.

## Motivation to overcome protective barrier?

A barrier test was used to test if maladaptive attraction was appetitively motivated, in the sense of being willing to seek out and exert flexible effort to overcome a protective barrier to reach the shock rod. A subset of rats that previously experienced shock rod tests (hSyn = 17, D1 = 13, eYFP = 8) were tested in a subsequent shock rod session, but with a shoebox-sized opaque barrier stretching across the center of the chamber, which prevented a rat from seeing or touching the shock rod (37 cm length × 13 cm width × 13 cm height, cardboard box wrapped in duct tape). A rat was initially placed on the far side of the barrier from the shock rod, so the barrier completely occluded the shock rod from view, unless the rat stood on hindlegs. The rat had to climb over the top of the barrier to reach the rod, and if so, laser was illuminated whenever the rat was within 2 cm proximity to the shock rod. If a rat crossed and touched the shock rod, the experimenter gently lifted the rat by hand within 10 s and returned it to the 'safe side' of the barrier opposite to the shock rod. If the rat climbed over the barrier again, this procedure was repeated for the duration of the 20 min session or until the rat reached the maximum 20 shocked contacts, whichever came first.

## Can shock cues become 'wanted'? Instrumental conditioned reinforcement test & odor preference test

Reward cues (Pavlovian CS+s) attributed with incentive salience can become attractive when encountered (e.g. CS+s for reward UCS may elicit sign-tracking approach and nibbling). Reward CS+s also may be sought out when absent (e.g., as assessed via instrumental conditioned reinforcement tests that earn CS+ presentations). To assess whether rats would seek shock rod-related CS+s as incentive cues, beyond shock rod attraction itself, we tested 1) a conditioned reinforcement test of whether rats would work to earn presentations of an auditory CS+ cue that had been associatively paired with voluntary shocks from the shock rod (i.e., co-activated whenever laser was illuminated), and 2) a preference/avoidance test of whether rats would seek out an olfactory contextual CS+ scent that was previously encountered under the shock rod and so was a contextual CS+ for shock rod encounters.

**Auditory CS+: conditioned reinforcement test.** We tested whether rats attracted to the shock rod would learn to perform a new instrumental nosepoke response to earn brief presentations of an auditory CS+ for shock alone (either pure tone or white noise, balanced across rats), without the presence of the shock rod, in a conditioned reinforcement test conducted in a separate MedAssociates chamber. Rats were presented with two fixed portholes in a 30 min session on each of two separate days. On one day, nose-pokes into a designated porthole earned a 4-s presentation of the auditory CS+ that previously had been paired with shock rod encounters (FR1). Nose-pokes into a second porthole produced nothing, and served to assess generalization and exploration. On a different day (order counterbalanced across rats), rats could earn 4-s presentations of an equally familiar CS− safe sound that they had previously heard an equal number of times in their home cage (white noise or tone, whichever had not been assigned as CS+), while pokes in the other porthole again earned nothing. The number of nose-pokes in each porthole was recorded.

**Contextual odor CS+ preference test.** A 2-chamber odor preference test was used to assess relative attraction to a contextual odor CS+ that had been previously associated with the shock rod context. On previous shock rod encounters, the CS+ odor always had been dabbed directly beneath the shock rod (either vanilla or orange extract, balanced across rats). A second CS− safe odor had been encountered an equal number of times/duration in the rat's home cage (whichever vanilla or orange scent was not CS+ for that rat). Finally, in an odor preference test, one chamber of a 2-chamber box was scented with the CS+ odor, and the other

chamber was scented with the CS− odor, and the rat was allowed to move freely. Time spent in each chamber during a 15 min test was recorded.

## Laser self-stimulation

To test if CeA ChR2 laser stimulation carried strong incentive value on its own, rats were tested for laser self-stimulation in an active instrumental response self-stimulation test in a different chamber (with no shock rod, dummy rod or electric shocks). Rats could earn brief 1-s CeA illuminations by either making instrumental nosepokes into one of two portholes, or for different rats by making instrumental contacts on one of two empty metal drinking spouts. For rats in the nose-poke test, one of two 2 cm portholes, spaced 7 cm apart on one chamber wall, was designated to deliver laser. The other porthole was designated as inactive (no-laser) to assess baseline level of exploratory nosepokes (porthole assignment counterbalanced across rats). Beam-breaks in the laser porthole earned 1 s of laser illumination controlled by Med PC software (fixed-ratio 1; 473 nm, 10 mW 40 Hz or 3 mW 25 Hz). For rats in the spout-touch task, each touch on one designated empty spout (laser spout) similarly earned 1 s laser illumination (fixed-ratio 1; 473 nm, 10 mW 40 Hz or 3 mW 25 Hz), whereas touches on another spout earned nothing. Self-stimulation tests were conducted in 3 daily 30 min sessions.

## Histological analyses of virus expression and Fos quantification

Fos expression in various brain structures was assessed to map recruitment of distant brain circuitry into activation during laser-paired shock rod encounters, as potential candidates to mediate shock rod attraction. Approximately 75 min prior to euthanasia, groups of hSyn ChR2, D1 ChR2, and eYFP control rats underwent a final 20 min session of shock rod encounter with shocks and laser administered as usual when the rat was within 2 cm of the shock rod (473 nm, 3 mW, 25 Hz). Another group of naïve control rats, which never had surgery or behavioral experimentation, were taken directly from their home cages for euthanasia and quantification of baseline brain levels of Fos activity.

Rats were deeply anesthetized 75 min after the onset of the above conditions with an overdose of sodium pentobarbital (150–200 mg/kg), and transcardially perfused with 0.1 M sodium phosphate buffer (PBS) followed by 4% paraformaldehyde (PFA). Brains were removed and post-fixed for 24 h in 4% PFA, then transferred to 25% sucrose PBS for cryoprotection, and coronally sectioned at 40 μm using a cryostat (Leica). For Fos immunohistochemistry, brain sections were blocked in 5% normal donkey serum in 0.2% triton-X PBS for 60 min and incubated in polyclonal rabbit anti-cfos primary antibody (1:2500, Synaptic Systems) overnight for up to 24 h. Brain sections were then incubated in biotin-SP donkey anti-rabbit secondary antibody (1:300, Jackson Immuno) for 2 h, followed by incubation in Cy3 streptavidin (1:300, Jackson Immuno) for 90 min. Sections were then mounted, air dried, and cover-slipped in anti-fade Pro-long gold (Cell Signaling Technology). Images were taken at 10× magnification using a Leica epifluorescent microscope with excitation bands 490–510 and 515–545 to visualize virus eYFP and Fos expression, respectively.

To determine the spread of laser-induced ChR2 activation in CeA Fos plumes, the number of Fos+ cells were counted in fifteen successive blocks (50 ×50 μm) along eight radial arms emanating from a central point placed 300 μm directly below the optic fiber tip. Counting continued along each arm until two consecutive blocks containing zero Fos+ cells were reached. This point marked the radius of that arm. The cumulative number of Fos+ cells was calculated for each block emanating from the center up to the radius for each arm. Fos elevation in ChR2 rats was calculated as a percent change from corresponding blocks in eYFP rats, denoted as >200% or >300% elevation from baseline. Blocks with >200% elevation were connected between each arm to form elevated Fos plumes, while blocks with >300% elevation were connected to form plumes with even higher elevation.

## Fos quantification in distributed brain circuitry

Images of whole brain sections were used to count Fos expression in multiple structures and subregions: including anterior, middle, and posterior regions of the orbitofrontal cortex, insula, prelimbic cortex, infralimbic

cortex, nucleus accumbens core and shell, dorsal neostriatum, ventral pallidum, bed nucleus of the stria terminalis, amygdala, substantia nigra, and ventral tegmental area. For Fos quantification in each structure, 3 boxes (100 μm × 100 μm) were placed at standardized locations within each structure or subregion, and Fos expressing cells were counted from each box by researchers blind to the experimental conditions. Box placements were made in reference to the Paxinos and Watson rat brain atlas[35].

## Statistics and reproducibility
IBM SPSS software was used to perform all statistical analyses. ANOVAs, two-sided *t*-tests, and Pearson's correlations were used to analyze within-group effects and between-group effects. ANOVAs with significant interaction effects were followed by parametric paired-wise comparisons and independent *t*-tests to analyze post hoc comparisons using the Bonferroni correction. All tests used a confidence interval of 95% with two-tailed significance level of $p < 0.05$.

## Reporting summary
Further information on research design is available in the Nature Portfolio Reporting Summary linked to this article.

## Data availability
The data that support the findings of this study (Figs. 1–10, Supplementary Notes 1–3, and Supplementary Figs. 1–6) are made available through figshare repository https://doi.org/10.6084/m9.figshare.30055336. All other data are available from the corresponding author.

## Code availability
MedPC software code for the laser self-stimulation and auditory conditioned reinforcement tasks are made available through figshare repository https://doi.org/10.6084/m9.figshare.30057832.

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

## Acknowledgements

We give special thanks to Megan Hill, Isabelle Anderson, Lilly Rosenberg, and Emily Bok for their contributions and dedication to this study, along with the many other research students who have helped at various stages. We are also grateful to Emily Henson for performing the in situ hybridization. This work was supported by National Institutes of Health grants MH063649, DA015188, and T32DA7268.

## Author contributions

D.N. designed, collected, analyzed data, and wrote the paper. K.C.B. designed, analyzed data, and wrote the paper.

## Competing interests

The authors declare no competing interests.
