## [Transparent Peer Review file · Communications Biology]

Wanting what hurts: D1 dopamine neuronal stimulation in CeA is sufficient to induce maladaptive attraction

Corresponding Author: Dr David Nguyen

Version 0:

Reviewer comments:

Reviewer #1

(Remarks to the Author)

The paper entitled "Wanting what hurts: D1 dopamine receptor neuronal stimulation in central nucleus of amygdala is sufficient to induce maladaptive attraction to a shock rod" has been reviewed. The authors report the critical role of the CeA in attributing motivational salience to stimuli, even when such stimuli are aversive. The findings align with the incentive sensitization theory of addiction, wherein incentive "wants" can become uncoupled from hedonic "likes." By demonstrating the specific involvement of D1-expressing CeA neurons, the study advances understanding of the neurobiological underpinnings of maladaptive behaviors.

This paper addresses an important question about the role of CeA D1 dopamine receptor neurons in maladaptive attraction behaviors, contributing to our understanding of incentive salience in addiction-like conditions. However, several experimental and interpretive concerns significantly limit the clarity and robustness of the conclusions.

1. The data presented do not convincingly differentiate the role of CeA D1R neurons from general neuronal targeting via hSyn ChR2. Both groups (D1 ChR2 and hSyn ChR2) display highly similar patterns in shock rod attraction across behavioral and cellular outcomes. Since hSyn ChR2 indiscriminately targets most CeA neurons, the authors should explicitly highlight the key findings that establish D1R neurons as critical for generating maladaptive "wanting what hurts."
2. The shock parameters (amplitude: 0.2–0.5 mA; duration: $<0.25\text{--}>1$ s) are inconsistent, which raises concerns about uncontrolled variability. The authors should clarify whether this variability was random or systematic. Moreover, consistent shock parameters should be used in future experiments, and attraction data across different shock intensities would provide valuable insights.
3. To rule out the possibility that increased shock rod contact is due to reduced pain perception caused by light stimulation, it would be beneficial to conduct experiments assessing pain sensation in hSyn ChR2, D1 ChR2, A2a-Cre, and CRH-Cre rats. While the discussion claims that light stimulation does not modulate pain based on immediate avoidance and burying behaviors after stimulation cessation, the evidence provided is insufficient to support this conclusion definitively.
4. It is essential to provide baseline aversion data to shocks across groups. This data is critical to demonstrate that D1 activation converts the aversive shock into an attractive stimulus in the laser-paired shock rod tests.
5. Although optogenetics is used to manipulate CeA neurons, the study does not provide direct evidence showing that D1, D2, or CRH neurons are activated by light stimulation.
6. The claim that "D1 ChR2 and hSyn ChR2 rats show little defensive behavior" lacks supportive defensive behavior data in the figures.
7. In Figure 4, the results for hSyn ChR2 rats show similar nose-poking frequencies for the active and inactive portholes on the CS+ day (active: 6.5 ± 1.13 , inactive: 6.5 ± 1.21), which suggests no preference for CS+ rather than an inability to discriminate between the ports. Additional training is required to ensure the rats can distinguish between the two ports before conducting the experiment. Moreover, simply observing more total pokes on the CS+ day than on the CS- day is insufficient to claim that the CS+ shock-associated sound is "highly motivating" for hSyn ChR2 rats. Furthermore, the increased poking in the active porthole on the CS- day suggests a preference for the CS-, which also requires explanation. The experimental design of this behavioral test may need revision to better support the study's claims. Similarly, for D1 ChR2 rats, the observed higher poking for the inactive porthole (20 ± 6.3) than the CS+ porthole (14.67 ± 4.02) on the CS+ day makes it even harder to conclude that CS+ motivation increased compared to hSyn ChR2 rats.
8. While quantification graphs for c-Fos-positive cells are presented, representative images of c-Fos expression in the respective brain regions should also be included. Additionally, the authors should specify the exact locations within each brain region (e.g., bregma coordinates) used for quantification.

9. The rationale for using varying light stimulation parameters (e.g., 473 nm at 3 or 10 mW and 25 or 40 Hz) across behavioral experiments should be clearly explained. Behavioral data for each condition should be presented to account for potential differences in stimulation effects.
For D1 neurons, how does optic stimulation affect basal locomotion? Presenting basic locomotion data for each neuronal population under activation would help interpret the experiments.
10. Descriptions often lack corresponding figures. For example, lines 263–276 describe significant differences in time spent near the shock rod (<2 cm) between hSyn ChR2 and D1 ChR2 rats compared to eYFP controls, but no data or heatmaps are provided to support this claim.
11. Schematic 1 should include histology images for all groups (hSyn, D1, D2 [A2a], CRH, and eYFP). The images should include clear labeling, scale bars, and higher magnification views of the CeA.
12. The sample size disparity across groups (e.g., fewer A2a-Cre and CRH-Cre rats) weakens the reliability of statistical comparisons. A more balanced experimental design is needed to draw robust conclusions.
13. The use of different genetic backgrounds for WT and transgenic rats (e.g., Sprague-Dawley for WT, Long Evans for D1-Cre and A2-Cre, Wistar for CRH-Cre) may introduce variability in behavioral and sensory phenotypes. The authors should justify this choice and confirm that these differences do not affect the results.
14. The inclusion of both male and female rats is a strength; however, the absence of sex-specific analyses is a significant oversight.
15. Please provide more information (official name of line, number of line) about the Cre-rat line used in experiments
16. In line 74–75, correct “cre” to “Cre” and “crh” to “Crh.”

Reviewer #2

(Remarks to the Author)

In this manuscript, the authors explore whether a normally aversive stimulus, a shock rod, can become appetitive if paired with optogenetic stimulation of various cell types within the central nucleus of the amygdala. They first show that stim of either D1-receptor neurons or general stim produces significant engagement with the shock rod, in contrast to other cell types. The animals will also work to gain access to the rod by climbing over a barrier, and are similarly motivated by cues associated with the rod/stim. They attempt to show that a non-shocking rod, or general operant self-stim of these neurons does not show the same effect, with mixed results. Lastly, they show that shock/stim induces cFos in a variety of downstream nuclei. They conclude that this is an example of maladaptive attraction, and may share similar neurobiological underpinnings to addiction.

The manuscript is generally well-written, and there certainly does seem to be appetitive salience attributed to a typically negative reinforcer. Whether this is really incentive salience, or just the outcome of pairing of a somewhat negative reinforcer to a much more positive reinforcer (the stim), or even something else, is unclear. There are issues that make interpretation difficult. The first is that stimulation of the CeA, which is heavily implicated in analgesia, might create a confounding analgesic state that could render the shock appetitive by itself. Second, methodological differences between the shock and no-shock rod are confounding and do not allow for comparison, as currently described. Finally, the authors do not appear to statistically test whether operant self-stim is different between hSyn, D1, and eYFP rats (it appears to be). This suggests that the light/stim is reinforcing, and that the shock simply serves as a salient cue for this. There is a precedent for self-stimulation of CeA as reinforcing (Tonagawa and Bruchas labs, full citations below), including a publication that demonstrates hSyn-ChR2 expression in rat CeA produces empty spout licking and nose poking for light (Janak lab), which is an identical design. These citations are not included in this manuscript. Therefore, this manuscript needs to be reinterpreted in light of their own, and others' data, with a more parsimonious explanation that rats will engage with a negative reinforcer in order to earn a positive reinforcer, something the preclinical addiction literature has demonstrated in depth.

1) A key word used throughout this manuscript is “maladaptive”, but it’s not clear from this data whether that term is accurate. Would any operant responding to optogenetic stimulation be maladaptive? Or conversely, what if touching the shock rod allowed access to a sexually receptive female? Would that be maladaptive? Simply because the animals touch a shock rod does not make a situation maladaptive, especially since it appears that rats will operantly press for light (Figure 5, Fraser et al). Paradoxical engagement may be a better term.

2) The CeA is a critical nucleus in the control of pain, largely through ascending information from the PBN. Thus, stimulation of this area may either increase or decrease pain responses, depending on the cell type. This is very important because presumably the shock is painful/aversive, but there could be other elements of the stimulus that are actually appetitive. If the aversive elements are blunted, then the shock itself may switch to becoming appetitive. Have the authors tested whether stimulation of D1 neurons or the CeA in general (hSyn) promotes analgesia or hyperalgesia?

3) It is unclear from the methods why animals in the no-shock condition received one 20 min habituation session prior to two 20 min optogenetic stimulation sessions, as opposed to 3-opto paired sessions with the shock group. This initial habituation session would introduce latent inhibition to the rat, as they are learning that exploration of the non-shock rod does not result in anything. Thus, they are less likely to make the association during the subsequent 2 stimulation sessions. Did rats undergo the same habituation with the shock bar? Why are the number of sessions of optogenetic stimulation different? It seems very difficult to conclude anything from this experiment, since the experimental design is fundamentally different, at least as currently written.

Further, there are 2 values for both the intensity and frequency of the light delivered, as well as the shock (lines 103-107) for the shock group. This contrasts with the non-shock group that only received the lower power and frequency. Why are there 2 different frequencies and intensities for the shock group? What does the data look like when only one or the other is

included? If they are different, then the groups need to be broken out. Again, this makes interpretation very difficult because the variables used are not the same.

4) In figure 5, it does appear that there are significantly more responders and a higher mean for operant responding for hSyn and D1 than eYFP. However, this comparison is not present in the results. Instead, the authors only discuss results within each group and correlations to their shock-rod scores. But this across-group comparison is important, because if there is significantly more responding, then the stimulation itself is statistically reinforcing, and thus the shock may simply serve as a cue. Also, again there are 2 intensities and 2 frequencies of stimulation. Why is this, and are there differences?

These animals appear to be correlated to their own shock-rod scores, which means they have been put through multiple learning contingencies. What are the experimental timelines for these animals? Do they all go through shock-rod and instrumental training? What about the dummy rod animals, and odor tests? Without separate groups of animals for various contingencies, there could be an additional source of confounding here.

5) With some of the distributions in the data well-described as having skew, why are normal statistical tests used?

6) The fact that D1 targeted rats show similar levels of extinction responding to controls and non-responding groups would suggest this is not an addiction-like behavior, at least in this group.

7) Many of the heading titles (eg "Motivated rod attraction overcomes obstacle") need to be made more clear. Using complete sentences is fine for these.

8) For the cFos studies, were operant self-stim animals, or the non-shock rod rats taken for Fos as well? Because shock and stim may each contribute to Fos, it would seem important to include the relevant control groups.

Citations regarding operant self-stim for light in the CeA:

Fraser KM, Kim TH, Castro M, Drieu C, Padovan-Hernandez Y, Chen B, Pat F, Ottenheimer DJ, Janak PH. Encoding and context-dependent control of reward consumption within the central nucleus of the amygdala. *iScience*. 2024 Apr 1;27(5):109652. doi: 10.1016/j.isci.2024.109652. PMID: 38650988; PMCID: PMC11033178.

Seo DO, Funderburk SC, Bhatti DL, Motard LE, Newbold D, Girven KS, McCall JG, Krashes M, Sparta DR, Bruchas MR. A GABAergic Projection from the Centromedial Nuclei of the Amygdala to Ventromedial Prefrontal Cortex Modulates Reward Behavior. *J Neurosci*. 2016 Oct 19;36(42):10831-10842. doi: 10.1523/JNEUROSCI.1164-16.2016. PMID: 27798138; PMCID: PMC5083011.

Kim J, Zhang X, Muralidhar S, LeBlanc SA, Tonegawa S. Basolateral to Central Amygdala Neural Circuits for Appetitive Behaviors. *Neuron*. 2017 Mar 22;93(6):1464-1479.e5. doi: 10.1016/j.neuron.2017.02.034. PMID: 28334609; PMCID: PMC5480398.

Reviewer #3

(Remarks to the Author)

The manuscript by David Nguyen and Kent Berridge investigates the role of specific neuronal subpopulations in the central nucleus of the amygdala (CeA) in inducing maladaptive attraction to noxious stimuli, using optogenetic approaches in transgenic rats. It identifies D1 dopamine receptor-expressing neurons as key contributors to this behavior, drawing parallels with broader mechanisms of incentive motivation. The work builds upon previous findings, extending the understanding of neural circuits underlying maladaptive behaviors, with implications for addiction research. The identification of specific neuronal subpopulations provides an important advancement in understanding CeA functionality.

Despite its relevance, some concerns/doubts have been raised while reviewing the manuscript:

1. In the introduction section, it is mentioned that D1-expressing neurons co-express somatostatin. Has overlap of expression of the other markers – CRF, A2A – been reported? Is there for example co-expression of D1 and D2/A2A as it occurs in the striatum in a sub-set of neurons? How exclusive are these neuronal populations?

2. In the methods section, it is not clear how proximity to the shock rod was measured? Please elaborate on how this was measured in real-time and how was optical stimulation triggered?

3. Regarding the "Motivation to overcome occluding barrier" test, have other heights been tested? 13cm for an adult rat may not represent a significant level of difficulty.

4. Histology for all the animals used in the experiment should be shown in supplement. In addition, has overlap between Chr2 expression and expression of D1, A2A and CRF been done (e.g. RNAScope)? I am curious regarding the similarity in behavior between SYN Chr2 and D1 Chr2 mice – could this be because Chr2 expression in SYN mice is more restricted to D1+ neurons in the CeA?

5. The manuscript is dense, particularly in the Results section, with complex statistical details that may overwhelm readers.

Reducing the number of subheadings and summarizing the results could improve readability of the manuscript.

6. Please consider removing all the averages and standard deviations from the results section. Statistics and averages should not be repeated – authors should opt by either presenting them in results' description section or in Figure legends.

7. Schematic 1 should be included in Figure 1.

8. Between groups statistics should be present in panel b of figure 1.

9. In the discussion section, authors touch on valence plasticity and opposite effects under different conditions but do not sufficiently contextualize these findings within broader amygdala-related fear/reward literature.

Overall, this is a well-conceived and methodologically rigorous study. With revisions to enhance readability, clarity and contextual depth the manuscript will make a valuable contribution to the field.

Version 1:

Reviewer comments:

Reviewer #1

(Remarks to the Author)

The authors have adequately addressed my concerns. I do not have additional questions.

Reviewer #2

(Remarks to the Author)

The authors were relatively receptive to many of my, and the other reviewers comments, especially regarding the Fos data, and this certainly strengthens the manuscript. However, I still feel the data that is included in this manuscript still does not definitively provide evidence for a "maladaptive attraction". To illustrate this, one exercise would be to change the order of the figures, starting with figure 7, then 1, then 4.

Figure 7 demonstrates, I think relatively unambiguously, that rats in the hSyn and D1 groups will press for light. Absolutely there is spread in the data, but the fact remains that statistically, and with a large effect (means of ~250 reinforcers for hSyn, ~500 for D1, and 15 for eYFP, a ~15-fold difference), light is a reinforcer. If this was presented first, or by itself, the obvious conclusion is that light, delivered to the CeA, is reinforcing. And this has been published by others (Figure 4, Fraser et al) as previously commented. It's also important to consider that this is not a naïve cohort, as they already have shock/light experience, and this may explain why some animals do not press.

Now we look at Figure 1. Clearly there is a difference between hSyn, D1 and the other groups, but Figure 7 makes this interpretation very difficult, because the light has been established as reinforcing. The fact that there is no correlation is not important here, and really, these should have been separate groups of rats to cleanly test these two questions. If you show figure 7, you cannot then say that the light has no value, because it clearly does have value, and therefore, attraction to the shock is confounded with attraction to the light.

It is also important to note that the number of responses are still lower than in Figure 4 (below), so is this attraction, or simply a lack of avoidance? Are those the same thing?

Now, interestingly, in Figure 4, we do not see any obvious attraction to the non-shocking rod paired with light, compared to controls (Responses ~20 / session, which is notably still higher than the shock rod, but equivalent or lower than controls). This seems somewhat confusing given the Figure 7 results, but confounding issues are now present, including the fact that there was a first session where light was not delivered, and when it was delivered in sessions 2 and 3, it was only at 3mW. Either this group of rats needs to be given light on the first session (as with Figure 1), and at all powers, or animals from the shock design of Figure 1 need 1 day (at the beginning) with no light and just shocks. This is critically important, because prior experience with cue/outcome relationships shapes subsequent behavior to a large degree. The study design is not the same between the groups, and therefore no strong conclusion can be made.

I maintain that pain experiments should be performed, but this is at least discussed at length in the discussion.

In summary, if the authors want to publish as is, without performing the above experiments, they need to be far more conservative in their conclusions, and explicitly acknowledge that Figure 7 confounds interpretation of Figure 1, and that the study design precludes interpretation of figure 4 in relation to figure 1.

Reviewer #3

(Remarks to the Author)

I thank the authors for their efforts in addressing all my concerns in the manuscript review process. No further doubts have been raised upon analysis of the new version of the manuscript.

Best regards.

Version 2:

Reviewer comments:

Reviewer #2

(Remarks to the Author)

I appreciate the responses of the authors, and their willingness to perform some additional analyses, especially the percent attribution. This is an excellent way of merging the extant data with this, and I have no further issues.

We are grateful for the helpful comments and suggestions of the three reviewers. In this revision, we have aimed to incorporate their suggestions as well as address their criticisms and comments.

Our specific responses and revisions prompted by individual comments of each reviewer are described below.

Reviewer #1 (Remarks to the Author):

The paper entitled “Wanting what hurts: D1 dopamine receptor neuronal stimulation in central nucleus of amygdala is sufficient to induce maladaptive attraction to a shock rod ” has been reviewed. The authors report the critical role of the CeA in attributing motivational salience to stimuli, even when such stimuli are aversive. The findings align with the incentive sensitization theory of addiction, wherein incentive “wants” can become uncoupled from hedonic “likes.” By demonstrating the specific involvement of D1-expressing CeA neurons, the study advances understanding of the neurobiological underpinnings of maladaptive behaviors.

This paper addresses an important question about the role of CeA D1 dopamine receptor neurons in maladaptive attraction behaviors, contributing to our understanding of incentive salience in addiction-like conditions. However, several experimental and interpretive concerns significantly limit the clarity and robustness of the conclusions.

1. The data presented do not convincingly differentiate the role of CeA D1R neurons from general neuronal targeting via hSyn ChR2. Both groups (D1 ChR2 and hSyn ChR2) display highly similar patterns in shock rod attraction across behavioral and cellular outcomes. Since hSyn ChR2 indiscriminately targets most CeA neurons, the authors should explicitly highlight the key findings that establish D1R neurons as critical for generating maladaptive “wanting what hurts.”

Author’s response: We thank Reviewer 1 for the insightful comments regarding the similarity of hSyn ChR2 and D1 ChR2-specific effects. We have now tried to clarify the nature of our claim: we only conclude that paired selective stimulation of CeA D1 neurons is *sufficient* to generating shock rod attraction, essentially equivalent in magnitude to global hSyn stimulation of all CeA neurons. This shows that D1 neurons are especially capable of mediating the maladaptive attraction, by comparison to D2 neurons or CRF neurons that failed to induce attraction.

But we do not conclude that D1 neurons are *necessary* for shock rod attraction, since we are reporting a ‘gain of function’ study, in the form of *inducing* the maladaptive attraction via selective stimulation of particular CeA neuronal populations. A ‘loss of function’ claim of *necessity* would require a different manipulation, such as knocking out D1 neurons during hSyn

stimulation of all other CeA neurons. Possibly loss of CeA D1 neurons would prevent hSyn stimulation of other CeA neurons from inducing shock rod attraction. If so, that would indicate D1 neurons are necessary as well as sufficient. Alternatively, possibly the aggregate of all other remaining CeA neurons might be able to mediate some degree of shock rod attraction, even though selective stimulation at least of other D2 and CRF populations has so far failed to do so. If so, that would indicate D1 neurons are not strictly necessary, even though they are sufficient, to mediate shock rod attraction.

Regardless of the answer to the necessity question, we believe it is pressing is to identify whether any particular CeA neuronal population makes a special contribution to generating the maladaptive attraction, in the sense of being sufficient on its own, when selectively stimulated, to induce shock rod attraction. We have tried to make this clearer in the revised manuscript.

2. The shock parameters (amplitude: 0.2–0.5 mA; duration: <0.25–>1 s) are inconsistent, which raises concerns about uncontrolled variability. The authors should clarify whether this variability was random or systematic. Moreover, consistent shock parameters should be used in future experiments, and attraction data across different shock intensities would provide valuable insights.

Author's response: We thank Reviewer 1 for raising the issue of shock variability. Any variation in shock intensity here was due to *variation in the voluntary actions* rats used to make contact with the shock rod. Generated shock intensity was never varied, and was always set at 0.5 mA on the shock generator: that was the maximum shock received. Bare touches with skin, mouth or teeth were likely to receive full 0.5 mA. But protected touching with fur or nails reduced the measured shock amplitude. Similarly, duration of contact was a factor that also affected measured amplitudes: very brief contact durations of less than 0.5 sec reduced the intensity of rod shocks below maximum, whereas longer contacts lasting > 0.5 sec durations rose to the full 0.5 mA intensity. Based on our measurements, the received amplitude was never less than 0.2 mA and never higher than 0.5 mA. Most D1 ChR2 and hSyn ChR2 rats experienced a full range of these amplitudes as they made many contacts.

The second important point to note is that similar levels of shock rod attraction were produced in most D1 ChR2 and hSyn ChR2 rats despite any individual variation in shock intensity/duration produced by individual variation in voluntary contacts. This suggests that generation of shock rod attraction is a robust phenomenon that reliably emerges across a range of shock intensities between 0.2-0.5 mA.

3. To rule out the possibility that increased shock rod contact is due to reduced pain perception caused by light stimulation, it would be beneficial to conduct experiments assessing pain sensation in hSyn ChR2, D1 ChR2, A2a-Cre, and CRH-Cre rats. While the discussion claims that

light stimulation does not modulate pain based on immediate avoidance and burying behaviors after stimulation cessation, the evidence provided is insufficient to support this conclusion definitively.

Author's response: We agree that it is possible that CeA ChR2 stimulation may reduce pain perception, and have now restated that again in this revision. We previously had stated so in our lab's earlier publication on hSyn shock rod attraction (Warlow et al., 2020), but we neglected to mention in our original submission here. We now address it in the Discussion section of this revision.

However, we believe that pain reduction cannot fully explain CeA hSyn ChR2 and CeA D1 ChR2 shock rod attraction for several reasons. 1) First, the dummy rod, which was laser-paired but unelectrified and therefore gave least pain of all (zero) was less attractive than the shock rod. This suggests that laser seeking via rod attraction is not facilitated simply by removing pain. 2) Similarly, some CeA hSyn ChR2 and CeA D1 ChR2 rats that were attracted to the shock rod completely failed to self-stimulate laser by making nose pokes or touching an empty metal spout. That again indicates that laser-seeking via nonpainful routes is not the full explanation of shock rod attraction. 3) Finally, our previous article reported that CeA hSyn laser stimulation actually caused *increased* fearful freezing to an auditory CS+, and *increased* fearful avoidance of a contextual odor CS+ in a traditional Pavlovian fear conditioning paradigm with a painful footshock UCS (Warlow, 2020). Increased defensive responses to CS+s for shock is the opposite of what one would expect if CeA ChR2 stimulation reduced the painfulness of the shock UCS. That is if CeA stimulation reliably reduced UCS pain, one would expect reduced fear CRs rather than increased fear CRs. Our tentative explanation is that shock is unavoidable and uncontrollable in the Pavlovian fear conditioning situation, but is under voluntary control and is assignable to the shock rod here, while the rest of the chamber is safe. Overall, we believe that pairing CeA stimulation can increase the motivational salience of cues for paired affective UCSs, either increasing incentive salience in a controllable shock rod situation or increasing fearful salience in a Pavlovian fear conditioning situation. This is now briefly discussed in the Discussion section of this revision.

4. It is essential to provide baseline aversion data to shocks across groups. This data is critical to demonstrate that D1 activation converts the aversive shock into an attractive stimulus in the laser-paired shock rod tests.

Author's response: We suggest that between-subject baseline data demonstrating aversion to the painful shock rod is provided by the CeA eYFP control rats, which after one or two touches subsequently avoided the shock rod and sometimes employed defensive burying responses toward it. This baseline aversion to the shock rod was also confirmed by the D2 ChR2 rats and CRF ChR2 rats which failed to show shock rod attraction, and instead avoided and emitted defensive responses similarly to eYFP controls. Finally, within-subject aversion data was also

provided on Day 4 when laser was temporarily discontinued for D1 ChR2 rats and hSyn ChR2. Those rats then showed nearly immediate avoidance of the shock rod despite their previous attraction to the laser-paired shock rod on Days 1-3, and resumption of attraction on Day 5 when CeA laser stimulation was resumed.

5. Although optogenetics is used to manipulate CeA neurons, the study does not provide direct evidence showing that D1, D2, or CRH neurons are activated by light stimulation.

We thank the reviewer for raising this important point. We are glad to report that we have now added new data from an in situ hybridization analysis. We assessed whether *Cre* was specifically expressed particularly in D1 CeA neurons of D1 ChR2 rats that showed shock rod attraction. Messenger RNAs for *Cre* and D1 were visualized using fluorescence in situ hybridization, and co-localization of *Cre* mRNA and D1 mRNA in the same CeA neurons was confirmed. We now state in Results section that “*Cre* mRNA and D1 mRNA were observed to be colocalized together in the same CeA neurons in D1 ChR2 slices, amounting to approximately 40% of all CeA neurons counted (5.33 ± 1.45 colabeled neurons per $100 \times 100 \times 17 \mu\text{m}$ volume sample). Remaining CeA neurons contained neither *Cre* mRNA nor D1 mRNA. There were no neurons observed in D1 ChR2 slices expressing *Cre* but not D1 mRNAs, and none expressing D1 but not *Cre* mRNAs. Anatomically, *Cre*⁺ and D1⁺ coexpressing neurons were most densely concentrated in central and lateral subregions of CeA”.

6. The claim that “D1 ChR2 and hSyn ChR2 rats show little defensive behavior” lacks supportive defensive behavior data in the figures.

Author’s response: We have now added a graph showing defensive behavior and avoidance to figure 2g and figure 2h in this revision.

7. In Figure 4, the results for hSyn ChR2 rats show similar nose-poking frequencies for the active and inactive portholes on the CS+ day (active: 6.5 ± 1.13 , inactive: 6.5 ± 1.21), which suggests no preference for CS+ rather than an inability to discriminate between the ports. Additional training is required to ensure the rats can distinguish between the two ports before conducting the experiment. Moreover, simply observing more total pokes on the CS+ day than on the CS- day is insufficient to claim that the CS+ shock-associated sound is “highly motivating” for hSyn ChR2 rats. Furthermore, the increased poking in the active porthole on the CS- day suggests a preference for the CS-, which also requires explanation. The experimental design of this behavioral test may need revision to better support the study’s claims. Similarly, for D1 ChR2 rats, the observed higher poking for the inactive porthole (20 ± 6.3) than the CS+ porthole (14.67 ± 4.02) on the CS+ day makes it even harder to conclude that CS+ motivation increased compared to hSyn ChR2 rats.

Author's response: We are glad to provide further clarification on this issue. We have now added new additional eYFP control data that shows the shock-associated auditory CS+ had no incentive value for eYFP control rats, but did have positive incentive value for both D1 CeA ChR2 and hSyn CeA ChR2 rats that were attracted to the shock rod. That is, the CS+ did not boost instrumental nosepekes in eYFP rats, but did so in D1 ChR2 and in hSyn ChR2 rats in the instrumental conditioned reinforcement test.

To clarify further, it is important to emphasize that rats underwent *only one* conditioned reinforcement session with CS+, and one other session with CS-, with the order of sessions counterbalanced across rats. Only a single test with each auditory CS stimulus was used because these were essentially CS extinction trials: the CSs were presented multiple times in the absence of their previously associated UCSs (shock rod shocks vs home cage) during the trial. Therefore, the CS would be expected to substantially extinguish its association with UCS over the course of the session, and any subsequent session would be dealing with an already-extinguished CS. That meant that only a first test would be maximally informative. However, it appears from our data that the single session was not enough for the groups as a whole to learn a new instrumental discrimination association regarding the difference between active vs inactive noseports. Many rats appeared to generalize during the single sessions (which were their first and only exposure to the instrumental associations for CS+ on one day, and for CS- on the other day) between the porthole that triggered the auditory CS and the porthole that did not. For this reason, we believe the most informative comparison is between the *CS sessions*: that is, between the CS+ day (shock-associated sound) day vs the CS- day (safe sound). In particular, we point to the contrast between responding by eYFP control rats vs D1 ChR2 and hSyn ChR2 rats on those two days.

Our results show that both D1 ChR2 rats and hSyn ChR2 rats made twice as many nosepekes responses on the day they earned the CS+ than on the day they earned the CS-. This suggests the CS+ was more highly motivating than the CS- to these rats that had been attracted to the shock rod. By contrast, eYFP control rats did not differ in nosepekes between CS+ and CS- days, and eYFP rats were lower on both days than D1 ChR2 or hSyn ChR2 rats were on their CS+ day.

Finally, we note that our separate tests of the contextual olfactory CS+ vs CS- cues produced confirming evidence that shock CS+ cues gained positive incentive value for D1/hSyn ChR2 rats. One CS+ odor previously had been placed under the shock rod, and so was associated with shocks, whereas the other CS- odor was associated with the home cage. Rats were subsequently tested in a 2-chamber olfactory preference/avoidance apparatus. We found that D1 ChR2 and hSyn ChR2 rats both preferred to spend 150% more time in their shock-associated CS+ scented chamber than in the CS- scented chamber. Overall, we believe these conditioned reinforcement results support the conclusion that shock rod CS+s gained positive motivational value as incentive cues, for both D1 ChR2 rats and hSyn ChR2 rats that had been attracted to the laser paired shock rod.

8. *While quantification graphs for c-Fos-positive cells are presented, representative images of c-Fos expression in the respective brain regions should also be included. Additionally, the authors should specify the exact locations within each brain region (e.g., bregma coordinates) used for quantification.*

Author's response: This is an excellent suggestion, thank you. We have now added representative images for brain regions. We have also indicated appropriate stereotaxic coordinates for each region.

9. *The rationale for using varying light stimulation parameters (e.g., 473 nm at 3 or 10 mW and 25 or 40 Hz) across behavioral experiments should be clearly explained. Behavioral data for each condition should be presented to account for potential differences in stimulation effects.*

Author's response: We would have been concerned if it turned out that shock rod attraction was limited to only one particular stimulation configuration, as that would suggest attraction was an idiosyncratic artifact of a single stimulation condition. We therefore wished to know if shock rod attraction would remain robust across different CeA stimulation configurations. Our results indicate that shock rod attraction remained robust at similar intensities across 3 and 10 mW intensities and 25 and 40 Hz frequencies. Regarding particular parameter selection, many studies in the literature have used 10 mW intensity. But we believe that 10 mW is possibly high enough to generate heat that could modulate local neuronal function even in eYFP control rats (Tyssowski & Gray, 2019). For that reason, we also tested the lower 3mW intensity. Likewise for frequency, 40 Hz and 25 Hz have both been commonly used in the literature, and so both were assessed here. Our results showed that strong shock rod attraction was induced in D1 ChR2 and hSyn ChR2 rats regardless of which CeA stimulation configuration was used, and the level of attraction did not differ across configurations. We have now tried to clarify this rationale in Methods and Results, and have added a statistical analysis to results showing that these stimulation configurations did not differ in their intensity of maladaptive attraction.

10. *For D1 neurons, how does optic stimulation affect basal locomotion? Presenting basic locomotion data for each neuronal population under activation would help interpret the experiments.*

Author's response: We did not observe any changes in locomotion during CeA D1 ChR2 stimulation, possibly because rats received only brief CeA laser stimulations when they were voluntarily within 2 cm proximity to the shock rod. During laser stimulations, CeA D1 ChR2 rats typically remained right next to the rod while chewing on it or repeatedly touching it with paws or snout, rather than locomoting around the chamber. Also, even between laser stimulations, D1 ChR2 and hSyn ChR2 rats typically remained within 5-10 cm of the rod, returning to it a number of times.

11. Descriptions often lack corresponding figures. For example, lines 263–276 describe significant differences in time spent near the shock rod (<2 cm) between hSyn ChR2 and D1 ChR2 rats compared to eYFP controls, but no data or heatmaps are provided to support this claim.

Author's response: We now report the amounts of time spent in proximity to the shock rod. Basically, D1 ChR2 rats and hSyn ChR2 rats typically remained close to the shock rod even in between contacts, whereas eYFP control rats, D2 ChR2 rats and CRF ChR2 rats typically remained near a distant wall as far away as possible from the shock rod.

12. Schematic 1 should include histology images for all groups (hSyn, D1, D2 [A2a], CRH, and eYFP). The images should include clear labeling, scale bars, and higher magnification views of the CeA.

Author's response: Thank you for this excellent suggestion. We have now added histology images for the groups.

13. The sample size disparity across groups (e.g., fewer A2a-Cre and CRH-Cre rats) weakens the reliability of statistical comparisons. A more balanced experimental design is needed to draw robust conclusions.

Author's response: We should point out that no evidence of shock rod attraction was seen in any of our A2a-Cre rats or CRH-Cre rats. Not only statistically but also numerically their shock rod attraction was virtually identical to eYFP controls: rats from these groups typically made 1 or 2 initial touches of the shock rod in exploration, but virtually zero contacts afterwards and then emitted defensive burying or stayed as far away as possible. There was simply no evidence, even from any individual rats, that A2A or CRF stimulation induced any attraction at all.

14. The use of different genetic backgrounds for WT and transgenic rats (e.g., Sprague-Dawley for WT, Long Evans for D1-Cre and A2-Cre, Wistar for CRH-Cre) may introduce variability in behavioral and sensory phenotypes. The authors should justify this choice and confirm that these differences do not affect the results.

Author's response: The various transgenic rats unavoidably had different backgrounds. But rats from all strains were included in our control eYFP group, and our hSyn ChR2 wildtype group included rats from both Sprague-Dawley and Long Evans strains. Also, we suggest the comparable shock rod attraction induced in D1 ChR2 Long Evans rats and in hSyn Sprague-Dawley rats (as well as Long Evans) confirms that maladaptive attraction is a robust phenomenon, and not an artifact of one particular strain.

15. *The inclusion of both male and female rats is a strength; however, the absence of sex-specific analyses is a significant oversight.*

Author's response: We did statistically assess sex differences, but did not typically find any. We apologize that we neglected to report the negative results in our earlier version. We now do report that we found no difference between males and females in levels of shock rod attraction, barrier crossing to reach the shock rod, conditioned reinforcement, etc.

16. *Please provide more information (official name of line, number of line) about the Cre-rat line used in experiments*

In line 74–75, correct “cre” to “Cre” and “crh” to “Crh.”

Author's response: Thank you for pointing out these mistakes. They have now been corrected in the manuscript.

Reviewer #2 (Remarks to the Author):

In this manuscript, the authors explore whether a normally aversive stimulus, a shock rod, can become appetitive if paired with optogenetic stimulation of various cell types within the central nucleus of the amygdala. They first show that stim of either D1-receptor neurons or general stim produces significant engagement with the shock rod, in contrast to other cell types. The animals will also work to gain access to the rod by climbing over a barrier, and are similarly motivated by cues associated with the rod/stim. They attempt to show that a non-shocking rod, or general operant self-stim of these neurons does not show the same effect, with mixed results. Lastly, they show that shock/stim induces cFos in a variety of downstream nuclei. They conclude that this is an example of maladaptive attraction, and may share similar neurobiological underpinnings to addiction.

The manuscript is generally well-written, and there certainly does seem to be appetitive salience attributed to a typically negative reinforcer. Whether this is really incentive salience, or just the outcome of pairing of a somewhat negative reinforcer to a much more positive reinforcer (the stim), or even something else, is unclear. There are issues that make interpretation difficult. The first is that stimulation of the CeA, which is heavily implicated in analgesia, might create a confounding analgesic state that could render the shock appetitive by itself. Second, methodological differences between the shock and no-shock rod are confounding and do not allow for comparison, as currently described. Finally, the authors do not appear to statistically test whether operant self-stim is different between hSyn, D1, and eYFP rats (it appears to be). This suggests that the light/stim is reinforcing, and that the shock simply serves as a salient cue for this. There is a precedent for self-stimulation of CeA as reinforcing (Tonagawa and Bruchas

labs, full citations below), including a publication that demonstrates hSyn-ChR2 expression in rat CeA produces empty spout licking and nose poking for light (Janak lab), which is an identical design. These citations are not included in this manuscript. Therefore, this manuscript needs to be reinterpreted in light of their own, and others' data, with a more parsimonious explanation that rats will engage with a negative reinforcer in order to earn a positive reinforcer, something the preclinical addiction literature has demonstrated in depth.

We thank Reviewer 2 for the positive comments above about the demonstration of appetitive motivation for a normally-negative reinforcer. We also agree that CeA can support laser self-stimulation, and thank the reviewer for the citations which we have added to this revision. In our lab's earlier 2020 publication we did cite the Kim et al. 2017 self-stimulation paper, and we ourselves obtained significant self-stimulation both in our 2020 results and here. But it is necessary to realize that *all those significant positive self-stimulation reports are group effects*; they mask the fact that there is considerable individual variation in optogenetic CeA laser self-stimulation. Some studies only report effects for entire groups without graphing individuals within the group, but studies that also show individuals' performance reveal tremendous heterogeneity in CeA laser self-stimulation. Specifically, although some individuals self-stimulate laser robustly, other hSyn ChR2 and D1 ChR2 individuals fail to self-stimulate for CeA laser at all. For positive self-stimulators, some shock rod attraction might indeed be due to laser-seeking as the reviewer suggests. But what is important here is that we find equally strong shock-rod attraction both here and in our 2020 results even in CeA ChR2 rats that fail to self-stimulate laser at all when offered the chance to do so without shock by simply making a nosepoke, touching an empty metal spout, or pressing a lever. Overall, we find **no correlation** between *strength of shock-rod attraction (which is nearly universal in our hSyn ChR2 and D1 ChR2 rats)* and *strength of laser self-stimulation (which is shown robustly by some hSyn and D1 rats, mildly by other individuals, and not at all by another set of individuals)*. This is why we suggest that shock rod attraction is not fully explainable simply by seeking laser self-stimulation.

In our earlier papers and here in Discussion we have ventured a hypothesis to explain why CeA laser self-stimulation is weaker than laser-paired shock rod attraction (and weaker than attraction to laser paired cocaine or sucrose, as shown in earlier studies), suggesting that the limbic activation triggered by shock/sucrose/cocaine UCS interacts with the simultaneous CeA activation of ChR2 laser to assign maximal incentive salience to the target (more than produced by CeA laser alone). Whether that explanation is correct or not, we are sure that intense shock rod attraction can be induced even in D1/hSyn rats that fail to otherwise self-stimulate laser. We now discuss the issue more thoroughly in the Results and Discussion sections of this revision.

1) A key word used throughout this manuscript is "maladaptive", but it's not clear from this data whether that term is accurate. Would any operant responding to optogenetic stimulation be

maladaptive? Or conversely, what if touching the shock rod allowed access to a sexually receptive female? Would that be maladaptive? Simply because the animals touch a shock rod does not make a situation maladaptive, especially since it appears that rats will operantly press for light (Figure 5, Fraser et al). Paradoxical engagement may be a better term.

Author's response: We believe that shock rod attraction can legitimately be called maladaptive because of the following features: 1) The shock rod is noxious in the sense of eliciting flinches, avoidance, and defensive burying in eYFP control, D2 ChR2, and CRF ChR2 rats. 2) Shocks still elicit aversive flinches in D1 ChR2 and hSyn ChR2 rats that seek the rod and its cues. 3) The shock rod also elicits avoidance in D1 ChR2 and hSyn ChR2 as soon as the laser is turned off on Day 4. 4) Shock rod attraction cannot be fully explained by laser seeking as self-stimulation, as described above.

Overall, the crucial criteria for calling it maladaptive is that the target shock rod delivers a negative outcome of shocks, and delivers no detectable positive outcome (at least, for D1/hSyn rats that fail to self-stimulate laser in innocuous self-stimulation situations). If touching the shock rod delivered a sexual partner or any other reward, we would not call it maladaptive. We also would not call it maladaptive if all rats attracted to the shock rod also self-stimulated in nose poke, spout touch, lever press, etc. stimulations, or if individual shock-rod attraction correlated with individual self-stimulation scores, or if rats simply hovered closely over the rod to turn on laser without actually touching the rod. All those imaginary results are explainable by conventional reinforcement concepts. But those results were not observed. We suggest that shock rod attraction resists conventional explanations because it does not require a conventional reinforcer. That it delivers an outcome of solely noxious shocks makes it maladaptive in our view.

2) The CeA is a critical nucleus in the control of pain, largely through ascending information from the PBN. Thus, stimulation of this area may either increase or decrease pain responses, depending on the cell type. This is very important because presumably the shock is painful/aversive, but there could be other elements of the stimulus that are actually appetitive. If the aversive elements are blunted, then the shock itself may switch to becoming appetitive. Have the authors tested whether stimulation of D1 neurons or the CeA in general (hSyn) promotes analgesia or hyperalgesia?

Author's response: As discussed above for Reviewer 1, we don't deny that CeA ChR2 stimulation may possibly alter pain perception. However, we do observe that rod contacts are typically followed by a flinch response in all our rats. Also, we note 1) in our earlier study (Warlow et al., 2020), we reported that paired CeA ChR2 stimulation in a traditional Pavlovian fear conditioning paradigm increased **defensive CRs** of auditory CS+ elicited freezing and olfactory CS+ avoidance, even in the same CeA hSyn rats that were attracted to their laser-paired shock rod. Increased CS+-triggered fear conditioned reactions would be consistent with

increased pain, and is consistent also with increased motivational salience (our favored explanation), but is not consistent with the hypothesis that CeA stimulation reliably reduces shock pain perception.

3) It is unclear from the methods why animals in the no-shock condition received one 20 min habituation session prior to two 20 min optogenetic stimulation sessions, as opposed to 3-opto paired sessions with the shock group. This initial habituation session would introduce latent inhibition to the rat, as they are learning that exploration of the non-shock rod does not result in anything. Thus, they are less likely to make the association during the subsequent 2 stimulation sessions. Did rats undergo the same habituation with the shock bar? Why are the number of sessions of optogenetic stimulation different? It seems very difficult to conclude anything from this experiment, since the experimental design is fundamentally different, at least as currently written.

Author's response: The only conclusion we suggest that emerges from the dummy rod results is that laser-pairing does not make an innocuous dummy rod as strongly attractive as the shock rod. This is consistent with the relatively weaker laser self-stimulation performance compared to robust shock rod attraction.

In our previous study, we presented a dummy rod in 3 laser-pairings that were identical to the shock rod pairings, and failed to induce strong attraction to dummy rod. Here we wanted to pose an easier test for dummy rod attraction and simply see if laser-pairing made the dummy rod more attractive than it was without the laser. But the condition presented first could be confounded by novelty, unless a single habituation session was offered first in an A-B-A or A-A-B design, where A=No-laser and B=Laser-paired. We have acknowledged this limitation in the discussion section and have included an explanation of the rationale for the habituation session in the methods section. Overall, even though counterintuitive, we believe the laser-paired dummy rod never becomes as attractive as the laser-paired shock rod – regardless of which procedure we use.

Further, there are 2 values for both the intensity and frequency of the light delivered, as well as the shock (lines 103-107) for the shock group. This contrasts with the non-shock group that only received the lower power and frequency. Why are there 2 different frequencies and intensities for the shock group? What does the data look like when only one or the other is included? If they are different, then the groups need to be broken out. Again, this makes interpretation very difficult because the variables used are not the same.

Author's response: Regarding the range of shock intensities, as described above in reply to Reviewer 1, the shock amplitude was always set at 0.5 mA on our shock generator. Maximal intensity would be fully 0.5 mA if directly touched by mouth or by skin of paw. However, the actual shock received could be submaximal only briefly if touched by fur or nails, or for less than 0.5 sec duration, but the shock was never less than 0.2 mA according to our measurements. For

these reasons, we listed the range of shock amplitude as ~0.2 mA to 0.5 mA. We have now added a better explanation in the methods section to clarify this in the revision.

Regarding the range of laser parameters for shock rod, our primary aim was to assess whether CeA ChR2 shock rod attraction was a limited artifact of just one laser stimulation parameter. Alternatively, shock rod attraction could be a more robust CeA phenomenon that was reliably elicited across a range of stimulation parameters? Our results suggest the latter, indicating that shock rod attraction is robust across multiple CeA laser parameters.

4) In figure 5, it does appear that there are significantly more responders and a higher mean for operant responding for hSyn and D1 than eYFP. However, this comparison is not present in the results. Instead, the authors only discuss results within each group and correlations to their shock-rod scores. But this across-group comparison is important, because if there is significantly more responding, then the stimulation itself is statistically reinforcing, and thus the shock may simply serve as a cue. Also, again there are 2 intensities and 2 frequencies of stimulation. Why is this, and are there differences?

These animals appear to be correlated to their own shock-rod scores, which means they have been put through multiple learning contingencies. What are the experimental timelines for these animals? Do they all go through shock-rod and instrumental training? What about the dummy rod animals, and odor tests? Without separate groups of animals for various contingencies, there could be an additional source of confounding here.

Author's response: Yes, we confirm that D1 ChR2 and hSyn ChR2 rats showed higher group responses than eYFP, CRF or A2 groups. However, as described above, some D1 ChR2 and hSyn ChR2 individuals failed to self-stimulate at all. Yet shock rod attraction was as high even in those non-self-stimulating individuals as in those that did self-stimulate. For this reason, we conclude that, shock rod attraction is not completely reducible to a form of laser self-stimulation. If all individuals that were attracted to shock rod also self-stimulated laser in other innocuous settings, we would agree with the reviewer that the shock might function only as a cue or as a price to be paid for laser (although we also note that rats did not actually need to touch the rod to turn on laser, as long as they stayed within 2 cm proximity – yet they did touch and so received shocks that would be deemed unnecessary by the self-stimulation hypothesis).

We now state that the timeline for the experiments follows the order in which they are listed in the methods section. All rats began with the shock rod encounters to assess whether they were attracted. After all rats completed the shock rod tests, all hSyn and D1 rats that had shown shock rod attraction progressed to the barrier test, CS+ tests, dummy rod tests, and self-stimulation tests, along with a subset of other rats from those and from eYFP, D2, and CRF groups.

5) *With some of the distributions in the data well-described as having skew, why are normal statistical tests used?*

Author's response: This is a good point. It was primarily the laser self-stimulation tests that showed skew and strong individual differences, and we have now added nonparametric tests to that section. The results and conclusions remain unchanged.

6) *The fact that D1 targeted rats show similar levels of extinction responding to controls and non-responding groups would suggest this is not an addiction-like behavior, at least in this group.*

Author's response: Yes, we agree. Neither the D1 ChR2 rats nor hSyn ChR2 rats were permanently sensitized. Their attraction to the shock rod, which we believe is mediated by mesolimbic incentive salience or 'wanting', required the concurrent brain state of CeA D1 (or hSyn) optogenetic stimulation. Magnification of cue-triggered incentive salience above previously learned levels typically does require a facilitating physiological or brain state (e.g., drug priming; caloric hunger; sodium appetite; drug intracranial microinjection or optogenetic stimulation; mesolimbic sensitization; etc.) By comparison, addicted human individuals are posited to have an enduring brain state in the form of a sensitized mesolimbic dopamine-related system by the incentive sensitization theory of addiction, causing limbic hyper-reactivity to the addictive cues that trigger 'wanting'. Mesolimbic sensitization, once induced, is much more enduring than the temporary mesocorticolimbic activation induced by CeA optogenetic stimulation in nonsensitized rats.

We do not suggest that D1 or hSyn rats become permanently addicted to the shock rod in any sense. Rather, we suggest paired CeA ChR2 laser stimulation activated mesocorticolimbic circuitry to generate intense incentive salience underlying shock rod attraction, mimicking the excessive 'wanting' posited to occur due to more permanent incentive sensitization in addiction. Whether the reactive brain state is drug-induced mesolimbic sensitization or optogenetic CeA stimulation, the functional product of excessive 'wanting' that is detached from 'liking' may be similar although the time course is not.

7) *Many of the heading titles (e.g. "Motivated rod attraction overcomes obstacle") need to be made more clear. Using complete sentences is fine for these.*

It is true we phrased some heading titles in abbreviated fashion, similarly to a running head, but have now extended a number of heading titles as suggested.

8) *For the cFos studies, were operant self-stim animals, or the non-shock rod rats taken for Fos as well? Because shock and stim may each contribute to Fos, it would seem important to include the relevant control groups.*

Author's response: Non-shock rod attracted eYFP control rats and naïve unoperated control rats were also taken for Fos analysis, in addition to shock rod attracted D1 ChR2 and hSyn ChR2 rats. Our primary Fos question was whether shock rod attraction was mediated by recruiting patterns of increased mesocorticolimbic activation, over and above levels seen in eYFP control rats that avoided shock rod as well as in naïve unoperated control rats. Increased mesocorticolimbic recruitment of structures such as ventral tegmentum and nucleus accumbens was consistent with mediation by incentive salience. For the Fos experiment, each hSyn ChR2 or D1 ChR2 rat underwent a final shock rod session 70 minutes before cardiac perfusion, to allow peak Fos expression. Our control groups consisted of eYFP rats that underwent the same shock rod session 70 minutes prior to perfusion, but showed avoidance or defensive behavior, as well as a second naïve group of unoperated rats that did not undergo any activity prior to perfusion. In our previous study, we found that shock rod attraction evoked similar mesocorticolimbic Fos expression as in rats seeking laser-paired cocaine reward or laser-paired sucrose reward. We do not doubt that laser self-stimulation reward also would evoke Fos expression too, but this was not our central question here. The central question was whether 'wanting what hurts' was accompanied by Fos elevation patterns consistent with incentive motivation demonstrated in previous studies, and different from shock rod evoked defensive behavior, in eYFP rats. We believe the answer to that question is yes.

Citations regarding operant self-stim for light in the CeA:

Fraser KM, Kim TH, Castro M, Drieu C, Padovan-Hernandez Y, Chen B, Pat F, Ottenheimer DJ, Janak PH. Encoding and context-dependent control of reward consumption within the central nucleus of the amygdala. iScience. 2024 Apr 1;27(5):109652. doi: 10.1016/j.isci.2024.109652. PMID: 38650988; PMCID: PMC11033178.

Seo DO, Funderburk SC, Bhatti DL, Motard LE, Newbold D, Girven KS, McCall JG, Krashes M, Sparta DR, Bruchas MR. A GABAergic Projection from the Centromedial Nuclei of the Amygdala to Ventromedial Prefrontal Cortex Modulates Reward Behavior. J Neurosci. 2016 Oct 19;36(42):10831-10842. doi: 10.1523/JNEUROSCI.1164-16.2016. PMID: 27798138; PMCID: PMC5083011.

Kim J, Zhang X, Muralidhar S, LeBlanc SA, Tonegawa S. Basolateral to Central Amygdala Neural Circuits for Appetitive Behaviors. Neuron. 2017 Mar 22;93(6):1464-1479.e5. doi: 10.1016/j.neuron.2017.02.034. PMID: 28334609; PMCID: PMC5480398.

We are grateful for these citations, and have added them to our revision.

Reviewer #3 (Remarks to the Author):

The manuscript by David Nguyen and Kent Berridge investigates the role of specific neuronal subpopulations in the central nucleus of the amygdala (CeA) in inducing maladaptive attraction to noxious stimuli, using optogenetic approaches in transgenic rats. It identifies D1 dopamine receptor-expressing neurons as key contributors to this behavior, drawing parallels with broader mechanisms of incentive motivation. The work builds upon previous findings, extending the understanding of neural circuits underlying maladaptive behaviors, with implications for addiction research. The identification of specific neuronal subpopulations provides an important advancement in understanding CeA functionality.

Despite its relevance, some concerns/doubts have been raised while reviewing the manuscript:

1. In the introduction section, it is mentioned that D1-expressing neurons co-express somatostatin. Has overlap of expression of the other markers – CRF, A2A – been reported? Is there for example co-expression of D1 and D2/A2A as it occurs in the striatum in a sub-set of neurons? How exclusive are these neuronal populations?

We thank Reviewer 3 for the positive comments above. Regarding co-expression of neurotransmitters, yes, CeA CRF neurons do co-release a number of other neurotransmitters, including GABA, glutamate, dynorphin, and neurotensin. Regarding D1 receptor and D2 receptor segregation on different neurons vs co-expression of both receptors in the same CeA neuron, we believe the data of Kim et al. (2017) and others suggests D1 vs D2 receptors are largely segregated on different neurons in CeA, rather than be co-expressed. The CeA has been suggested to be a ‘striatal-type’ structure in macrosystem analysis, and segregation suggests that CeA distribution of D1 and D2 receptors is more similar to an exclusive dorsal neostriatum segregation pattern rather than a nucleus accumbens partial co-expression pattern. It is also worth mentioning that our stimulation results for D1 vs D2 neurons in CeA were opposite of each other: D1 stimulation produced strong shock rod attraction (and in some but not all individuals, also laser self-stimulation) whereas D2 stimulation produced no shock rod attraction (and some evidence for D2 laser avoidance). Different functional results of D1 stimulation vs D2 stimulation seems consistent with different neurons being activated.

2. In the methods section, it is not clear how proximity to the shock rod was measured? Please elaborate on how this was measured in real-time and how was optical stimulation triggered?

Author’s response: A trained experimenter continually monitored shock rod encounters and manually activated laser whenever a rat’s body part entered the 2 cm proximity zone, ceasing laser when the rat left the zone. We acknowledge that there is a margin of human error when making the judgement for where the 2 cm proximity zone begins during each shock rod approach, but believe laser activation remained closely paired with shock rod contacts. We have now noted this in the methods section.

3. Regarding the “Motivation to overcome occluding barrier” test, have other heights been tested? 13cm for an adult rat may not represent a significant level of difficulty.

Author’s response: We agree the barrier imposed only a mild degree of difficulty, and it was not meant to be an arduous task. We chose the 13 cm barrier because it provided visual safety from the shock rod if the rat viewed the shock rod as an aversive threat. The height visually occluded sight of the shock rod, adding to the sense of safety, unless the rat voluntarily stood upright on hind legs and peered over. We agree that crossing the barrier was relatively easy for a rat, needing only the mild effort to climb and raise its entire body higher than its head normally would be. We wanted to know whether the D1 ChR2 rats and hSyn ChR2 rats would repeatedly cross several times in the same session, demonstrating a persistent desire to seek the shock rod (they were gently returned by hand to the ‘safe side’ immediately each time they touched the rod). Here hSyn ChR2 and D1 ChR2 rats did so multiple times within a single 20 min session, touching the shock rod each time, consistent with an appetitive motivation to reach the shock rod.

It could be interesting to someday impose larger barriers or to use a progressive ratio breakpoint task as the price to reach the shock rod, as the Reviewer perhaps intends to suggest. However, imposing an increasing range of costs to reach the rod was beyond our current scope.

4. Histology for all the animals used in the experiment should be shown in supplement. In addition, has overlap between ChR2 expression and expression of D1, A2A and CRF been done (e.g. RNAScope)? I am curious regarding the similarity in behavior between SYN ChR2 and D1 ChR2 mice – could this be because ChR2 expression in SYN mice is more restricted to D1+ neurons in the CeA?

Author’s response: We have now added new mRNA data from an in situ hybridization analysis to the Results section, as mentioned above. These new data confirm that D1 mRNA and *Cre* mRNA was colocalized in the same CeA D1 neurons of D1 ChR2 rats, providing evidence that ChR2 laser stimulation specifically activated D1 neurons in D1 ChR2 rats. While it is unlikely that hSyn ChR2 stimulation specifically activated D1 neurons in particular, we do believe that D1 neurons were among the multiple CeA neuronal types stimulated in hSyn ChR2 rats. It seems quite possible that D1 neuronal activation was the chief contributor to shock rod attraction in both D1 ChR2 and hSyn ChR2 rats.

5. The manuscript is dense, particularly in the Results section, with complex statistical details that may overwhelm readers. Reducing the number of subheadings and summarizing the results could improve readability of the manuscript.

We recognize that the results are dense with data and statistics. We have tried to clarify, summarize and improve readability in this revision. We have tried to only use subheadings to help conceptually organize the different types of results.

6. Please consider removing all the averages and standard deviations from the results section. Statistics and averages should not be repeated – authors should opt by either presenting them in results' description section or in Figure legends.

We have now tried to reduce duplication of statistics and means in this revision.

7. Schematic 1 should be included in Figure 1.

We thank the Reviewer for this helpful suggestion.

8. Between groups statistics should be present in panel b of figure 1.

We have now added this.

9. In the discussion section, authors touch on valence plasticity and opposite effects under different conditions but do not sufficiently contextualize these findings within broader amygdala-related fear/reward literature.

We agree our discussion of these important issues is somewhat limited, but are afraid of making the manuscript too long. We have tried to improve the discussion further in this revision, but hope to tackle the issues more completely in a future review paper.

Overall, this is a well-conceived and methodologically rigorous study. With revisions to enhance readability, clarity and contextual depth the manuscript will make a valuable contribution to the field.

We are very grateful for this positive assessment, and hope our revision will be viewed as an improvement over the original manuscript.

General Conclusion:

Again, we are grateful to all Reviewers for their helpful comments. Their comments and suggestions spurred us to improve the manuscript in many ways. We hope this new revision will prove acceptable.

Sincerely,

David Nguyen and Kent Berridge

Articles cited:

- Tyssowski, K. M., & Gray, J. M. (2019). Blue light increases neuronal activity-regulated gene expression in the absence of optogenetic proteins. *ENeuro*, *6*(5).
<https://doi.org/10.1523/ENEURO.0085-19.2019>
- Warlow, S. M., Naffziger, E. E., & Berridge, K. C. (2020). The central amygdala recruits mesocorticolimbic circuitry for pursuit of reward or pain. *Nature Communications*, *11*(2716), 1–15. <https://doi.org/10.1038/s41467-020-16407-1>

We are very grateful that Reviewer 1 and Reviewer 3 found our previous revision acceptable. We are also grateful for the new helpful comments of Reviewer 2. In this new re-revision, we have aimed to incorporate Reviewer 2's suggestions, as described below.

Reviewer #2 (Remarks to the Author):

The authors were relatively receptive to many of my, and the other reviewers comments, especially regarding the Fos data, and this certainly strengthens the manuscript. However, I still feel the data that is included in this manuscript still does not definitively provide evidence for a “maladaptive attraction”. To illustrate this, one exercise would be to change the order of the figures, starting with figure 7, then 1, then 4.

Figure 7 demonstrates, I think relatively unambiguously, that rats in the hSyn and D1 groups will press for light. Absolutely there is spread in the data, but the fact remains that statistically, and with a large effect (means of ~250 reinforcers for hSyn, ~500 for D1, and 15 for eYFP, a ~15-fold difference), light is a reinforcer. If this was presented first, or by itself, the obvious conclusion is that light, delivered to the CeA, is reinforcing. And this has been published by others (Figure 4, Fraser et al) as previously commented. It's also important to consider that this is not a naïve cohort, as they already have shock/light experience, and this may explain why some animals do not press.

Now we look at Figure 1. Clearly there is a difference between hSyn, D1 and the other groups, but Figure 7 makes this interpretation very difficult, because the light has been established as reinforcing. The fact that there is no correlation is not important here, and really, these should have been separate groups of rats to cleanly test these two questions. If you show figure 7, you cannot then say that the light has no value, because it clearly does have value, and therefore, attraction to the shock is confounded with attraction to the light.

In summary, if the authors want to publish as is, without performing the above experiments, they need to be far more conservative in their conclusions, and explicitly acknowledge that Figure 7 confounds interpretation of Figure 1, and that the study design precludes interpretation of figure 4 in relation to figure 1.

Authors' response:

We have now accepted Reviewer 2's suggestion that we reorder the figures, and thank you for the suggestion. We now begin the Results section with the laser self-stimulation results figure from the nose-poke/spout-touch self-stimulation task (formerly Figure 7), as Rev 2 suggests. This figure replicates previously published conclusions that CeA ChR2 excitation supports laser self-stimulation for hSyn ChR2 and D1 ChR2 groups as a whole, but it also reveals enormous within-group variation in self-stimulation, which we believe is also true of many previous studies (including all in our own lab). That is, some hSyn ChR2 and D1 ChR2 individuals self-stimulate intensely (i.e., over 50 illuminations per session) or at least moderately (10 to 49 illuminations), but other individuals consistently fail to self-stimulate on any of the three test days.

To help unpack the implications of this variability, we now also depict self-stimulation data for these two subgroups Fig 1D, and we track the two subgroups separately in subsequent figures showing shock rod attraction (Figs 2 & 3), barrier crossing to reach shock rod (Fig 4), and dummy rod attraction (Fig 5). This allows explicit comparison of shock rod attraction in D1 ChR2 and hSyn ChR1 rats that failed to self-stimulate versus attraction in rats that did self-stimulate on at least one day at the moderate >10 criterion (bottom row shows the two groups separately for D1 ChR2 and hSyn ChR2 groups).

This new shock-rod figure shows that Reviewer 2 is indeed right that the propensity to self-stimulate CeA laser does contribute additional shock-rod attraction, and especially for D1 rats (Fig 2 f & h blue), and for hSyn rats at least on later days Fig 2 f & g orange). But the figure also shows that substantial shock rod attraction also occurs in D1 and hSyn rats that consistently failed to self-stimulate in the nose-poke/spout-touch test. For example, on Day 1 hSyn ChR2 non-self-stimulators showed as much shock rod attraction as hSyn ChR2 self-stimulators (Fig 2 f orange), and had a similar proportion of individuals that reached the 20-shock maximum and had to be removed early from the shock rod chamber (Fig 2 F orange). However, on hSyn ChR2 subsequent days 2 and 3, the Reviewer 2's point is supported: collapsing across all Days 1-3 shows that self-stimulation propensity added about 14% to hSyn ChR2 shock rod attraction (Fig 2 g), and added about 20% extra attraction for D1 ChR2 rats (Fig 2 h). Finally, on the Laser Reinstatement day (Day 5) conducted after the Laser Extinction Day (Day 4), a self-stimulation propensity added nearly 40% extra reinstatement of shock rod attraction for hSyn ChR2 rats (bottom right orange), and added extra 45% attraction for D1 ChR2 rats (bottom right blue). This certainly proves that Reviewer 2 is right that the motivation to self-stimulate can contribute to shock rod attraction.

Our new analysis helps quantify how much self-stimulation contributes to shock rod attraction. We now explicitly affirm that a self-stimulation propensity adds 15%-45%, especially for the maintenance across days and the reinstatement after laser extinction of shock rod attraction. However, we also note that initial induction of hSyn shock rod attraction on Day 1 is mostly independent of self-stimulation, and that at least some D1 ChR2 and hSyn ChR2 individuals that consistently failed to self-stimulate nonetheless reached the 20-shock maximum in shock rod attraction, especially on Day 1 but also even on later days. We suggest that for an individual that consistently refuses to self-stimulate when no shock is incurred to voluntarily receive 20 shocks from the shock rod in a single session is the best evidence that shock rod approach and contacts involve a degree of maladaptive attraction that is difficult to explain by a CeA ChR2 self-stimulation account.

Reviewer 2 continues: *It is also important to note that the number of responses are still lower than in Figure 4 (below), so is this attraction, or simply a lack of avoidance? Are those the same thing?*

Now, interestingly, in Figure 4, we do not see any obvious attraction to the non-shocking rod paired with light, compared to controls (Responses ~20 / session, which is notably still higher than the shock rod, but equivalent or lower than controls). This seems somewhat confusing given the Figure 7 results, but confounding issues are now present, including the fact that there was a first session where light was not delivered, and when it was delivered in sessions 2 and 3, it was only at 3mW. Either this group of rats needs to be given light on the first session (as with Figure 1), and at all powers, or animals from the shock design of Figure 1 need 1 day (at the beginning) with no light and just shocks. This is critically important, because prior experience with cue/outcome relationships shapes subsequent behavior to a large degree. The study design is not the same between the groups, and therefore no strong conclusion can be made. I maintain that pain experiments should be performed, but this is at least discussed at length in the discussion.

Authors' response: Yes, we agree it is also important to assess attraction to the laser-paired dummy rod that gave no shocks (Formerly Figure 4, now Figure 5). Similarly to our new shock rod figure, we now separately present dummy rod attraction in individuals that failed to self-stimulate CeA laser versus

individuals that did at least moderately self-stimulate >10 illuminations on any day for both D1 ChR2 and hSyn ChR2 groups (below, right). This comparison suggests that dummy rod contacts were 17% lower in hSyn ChR2 self-stimulators than non-self-stimulators, but 33% higher in D1 ChR2 self-stimulators than non-self-stimulators. We conclude that self-stimulation may contribute moderately to D1 ChR2 dummy rod contacts, but apparently not much to hSyn ChR2 dummy rod contacts.

We also agree with Reviewer 2 that shock rod contacts tend to be lower than dummy rod contacts, but note that shock rod contacts had a 20-maximum limit per session – as a rat was removed early if it received 20 shocks – whereas the dummy rod test had no contact limit. Having several individuals touch the dummy rod 30-40 times naturally tends to raise the mean number of dummy rod contacts higher than the mean shock rod contacts. Finally, we note that dummy rod contacts tend to be highest in eYFP control rats that had no ChR2 self-stimulation, suggesting most dummy rod contacts may be due to exploration, rather than self-stimulation.

Addition of new Limitations section: Finally, in agreement with Reviewer 2's and the Editor's concerns, we have now added a new Limitations section to the end of the Discussion to note several limitations of our study. These limitations include 1) *Dummy rod*: we note that our dummy rod procedure here was not identical to our shock rod procedure (although we note that our earlier study did use an identical dummy rod procedure, and found no dummy rod attraction: Warlow et al., 2020). We explain that here we wanted

to pose an easier test for dummy rod attraction, and simply see if the dummy rod became more attractive when it delivered laser than when it did not. However, results indicated it did not become much more attractive, though we believe it would be good for future studies to further explore both procedures. 2) *Experimental order*: We note that we tested for shock rod attraction before testing for self-stimulation here, because we wanted an uncontaminated assessment of initial shock rod attraction. If we ran nose-poke/spout-touch self-stimulation tests or dummy rod tests first, that could build up a positive valence association due to self-stimulation (in absence of shock) with CeA excitation that could then transfer to the shock rod. That would strengthen the self-stimulation account of shock rod attraction. Running the shock rod test first avoided that transfer effect. However, it would be valuable for future studies to vary the order of tests, and assess whether order alters shock rod attraction or self-stimulation. 3) *Self-stimulation contribution*: We agree with Reviewer 2 that it was necessary to further parse the role of CeA self-stimulation in shock rod attraction, and are grateful for your insistence that we do so. We note in Limitations that self-stimulation does contribute to maintenance of attraction across days and to reinstatement (and even on Day 1 for D1 ChR2 rats). However, we also note that some D1 ChR2 rats and hSyn ChR2 rats consistently failed to self-stimulate CeA laser, yet those failures-to-self-stimulate still showed substantial shock rod attraction (and in some instances reached the 20 shock maximum). We suggest that shock rod attraction specifically in individuals that otherwise fail to self-stimulate may be the best evidence that CeA-mediated shock rod attraction can involve a feature of maladaptive incentive salience.

Conclusion:

Again, we are grateful to Reviewer 2 for the helpful comments and suggestions. We believe the new changes have helped strengthen the manuscript. We hope this new revision will prove acceptable.

Sincerely,

David Nguyen and Kent Berridge